# Reframing Gaussian Splatting Densification with Complexity-Density Consistency of Primitives

**Zhemeng Dong, Junjun Jiang,** *Youyu Chen, Jiaxin Zhang, Kui Jiang, Xianming Liu**

Faculty of Computing, Harbin Institute of Technology

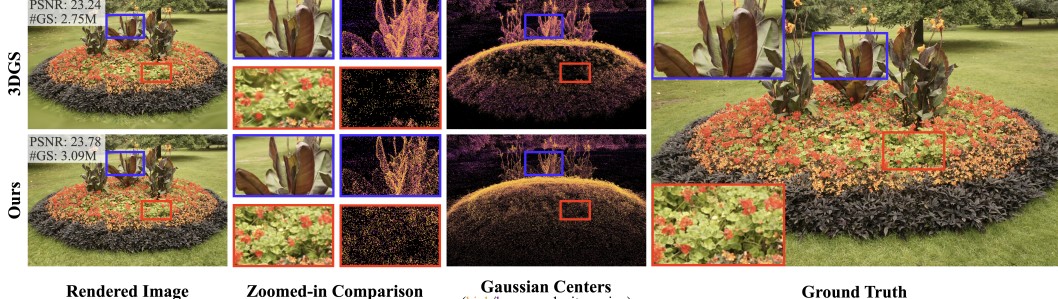

**Rendered Image**  **Zoomed-in Comparison**  **Gaussian Centers**  **Ground Truth**
(high/low complexity region)

Figure 1: Comparison between vanilla 3DGS and our CDC-GS on the rendering quality and distribution of Gaussian primitives. CDC-GS allocates Gaussian primitives in a more efficient fashion, using more primitives modeling complex region and less to model plain area, and thus significantly improves the rendering quality with a similar level of primitive number.

## Abstract

The essence of 3D Gaussian Splatting (3DGS) training is to smartly allocate Gaussian primitives, expressing complex regions with more primitives and vice versa. Prior researches typically mark out under-reconstructed regions in a rendering-loss-driven manner. However, such a loss-driven strategy is often dominated by low-frequency regions, which leads to insufficient modeling of high-frequency details in texture-rich regions. As a result, it yields a suboptimal spatial allocation of Gaussian primitives. This inspires us to excavate the loss-agnostic visual prior in training views to identify complex regions that need more primitives to model. Based on this insight, we propose Complexity-Density Consistent Gaussian Splatting (CDC-GS), which allocates primitives based on the consistency between visual complexity of training views and the density of primitives. Specifically, primitives involved in rendering high visual complexity areas are categorized as modeling high complexity regions, where we leverage the high frequency wavelet components of training views to measure the visual complexity. And the density of a primitive is computed with the inverse of geometric mean of its distance to the neighboring primitives. Guided by the positive correlation between primitive complexity and density, we determine primitives to be densified as well as pruned. Extensive experiments demonstrate that our CDC-GS surpasses the baseline methods in rendering quality by a large margin using the same amount of Gaussians. And we provide insightful analysis to reveal that our method serves perpendicularly to rendering loss in guiding Gaussian primitive allocation. Our implementations are available on our project page: cdc-gs.github.io.

---

*Corresponding author. E-mail: jiangjunjun@hit.edu.cn.

39th Conference on Neural Information Processing Systems (NeurIPS 2025).

# 1 Introduction

3D reconstruction, as a long investigated task of both computer vision and compute graphics communities, aims to build 3D representation of the scene from a batch of posed input views, which can be utilized for novel view synthesis, geometry reconstruction, semantic understanding and other downstream tasks. 3D Gaussian Splatting (3DGS) [1] has emerged as the leading paradigm for this task, offering a compelling combination of photorealistic and real-time rendering performance. Benefiting from explicitly modeling the scene with a set of Gaussian primitives, 3DGS is friendly to graphical rasterization pipeline, and thus being far more efficient than Neural Radiance Fields (NeRF) [2] in novel view synthesis. Recent researches have revealed that the allocation of Gaussian primitives is the key to high quality 3DGS representation [3, 4, 5, 6], referring to the densification and pruning of Gaussian primitives in 3DGS training process.

The training of 3DGS starts with a set of sparse Gaussian primitives, which are then iteratively optimized, densified, and pruned to form a well-distributed and compact representation. Vanilla 3DGS [1] assigns the positional gradient of each Gaussian primitive to themselves as the densification score, and mark out the coarse primitives with a densification score above the densification threshold to be densified. While recent works have explored alternative densification scores beyond positional gradient [7, 4, 5, 8], they are basically driven by rendering loss. Such loss-driven densification schemes typically struggle with expressing high-complexity visual details, where the rendering loss is smoothed out and fails to mark out the coarse Gaussian primitives for densification (Figure 1). Though one can solve this problem by decreasing the densification threshold for more primitives, this will lead to explosive growth of primitives in plain regions. These redundant primitives can barely improve the rendering quality but hinder the rendering efficiency.

Inspired by the fact that the failure of loss-driven densification happens in high visual complexity regions, we argue that we can specify these regions with visual prior measuring the visual complexity. Sparse Gaussian primitives in complex regions should be densified to allocate more primitives to the regions, while the redundant dense primitives in plain regions should be pruned for simplification and efficiency. With this insight, we propose Complexity-Density Consistent Gaussian Splatting (CDC-GS), which serves perpendicularly as a Gaussian primitive allocation method to existing loss-driven ones. Our method measures the visual complexity of training views with the high frequency component of Discrete Wavelet Transform (DWT) [9], and categorizes the primitives involved most with rendering the high frequency areas of training views as modeling complex regions. To measure the density of the neighboring space around a primitive, we utilize the inverse of geometric mean of its distance to the neighboring primitives. Combining the above two measurements, we define a loss-agnostic prior-based densification score, partitioning all primitives into four quadrants, where the two axes are complexity and density of primitives, as shown in Figure 2. Our CDC-GS suppresses the primitives in the left-up and right-down corner (see Figure 4a), which corresponds to the under-reconstructed and over-reconstructed regions, respectively.

To further improve the performance of CDC-GS, we introduce a complexity-aware adaptive densification threshold scheme, which adjusts the densification threshold in a primitive-wise manner. Primitives modeling complex regions are assigned with a lower densification threshold, which helps to allocate more Gaussians to complex regions, and vice versa.

Extensive experiments are performed on standard 3DGS benchmarks, which demonstrate that our method substantially improves the rendering quality with the same level of primitive number compared to baselines. We also provide insightful analysis on why the proposed prior-based primitive allocation strategy are perpendicular to existing loss-based ones, which further advocates the soundness of our method. To summarize our contributions,

- We propose a novel Gaussian primitive allocation scheme based on loss-agnostic visual prior, which guides 3DGS to allocate Gaussian primitives smartly by modeling complex regions with dense primitives and plain regions with sparse ones.

- We develop Complexity-Density Consistent Gaussian Splatting, which utilizes the wavelet frequency prior and primitive density to detect primitives in under-reconstructed and over-reconstructed regions for densification and pruning.

- Insightful analysis is provided to demonstrate why the proposed complexity-density consistent prior serves as a perpendicular primitive allocation strategy against existing loss-driven methods.

## 2 Related Work

**3D Reconstruction.** The target of 3D reconstruction is to obtain high quality 3D representation of the scene. Each time when the representation paradigm evolves always brings a new trend to the research community. Early works use point clouds to reconstruct the scene [10, 11, 12, 13], which remain popular these days in certain domains but struggle to recover textures, failing in novel view synthesis. Alternatively, Neural Radiance Fields (NeRF) [2] represents the scene as an implicit radiance field encoded in a multi-layer perceptron (MLP). While showing significant advantage in novel view synthesis against prior methods, NeRF based reconstruction methods have bottleneck in the efficiency of training and rendering [14, 15, 16, 17]. More recently, 3D Gaussian Splatting (3DGS) [1] emerges as a new leading paradigm for 3D reconstruction, highlighted with fast training and real-time rendering. Over time, numerous extensions have further enhanced its efficiency [8, 18, 19, 20], compactness [21, 22, 23, 24], and robustness under challenging conditions [25, 26, 27, 28, 29]. Using a set of 3D Gaussian primitives expressing the scene, 3DGS can be adopted to extensive tasks, such as novel view synthesis [30, 31, 32, 33], geometric reconstruction [34, 35, 36, 37, 38], 3D segmentation [39, 40, 41], and semantic understanding [42, 43, 44, 45].

**Primitive Allocation for 3DGS.** To obtain high quality 3DGS representation of the scene, it is vital to correctly allocate Gaussian primitives over the space. The allocation is expected to use more primitives to model the complex regions and less primitives to model the plain regions. For primitive densification, prior works propose different densification scores to measure if a primitive is modeling an under-reconstructed region. Vanilla 3DGS [1] adopts high positional gradient of primitives centers derived from the rendering loss as the densification score. Absgs [4] takes a step forward by calculating the sum of pixel-wise absolute gradient, addressing the conflicts in gradient direction of vanilla 3DGS [1]. Revising-3DGS [7] utilizes the primitive-involved high rendering error instead, which differs from positional gradient but is still driven by the rendering loss. Taming-3DGS [8] proposes a densification score mixing primitive attributes and positional gradient. All of these methods specifies under-reconstructed primitives in a loss-driven manner, thus they are either struggled with under-reconstruction in complex regions (vanilla 3DGS, Revising-3DGS, Taming-3DGS) or over-reconstruction in plain regions (Absgs). On the other hand, redundant primitives are pruned to slim the primitives. The most widely adopted metric is to prune primitives with low opacity [46, 47, 48]. There are also methods pruning the primitives with over-dense neighbors [3, 21].

**Frequency-based 3DGS Densification.** Since our work use frequency information of training views to guide the allocation of Gaussian primitives, we specifically introduce the works investigating the use of frequency theory for primitive allocation. FreGS [49] improves 3DGS optimization with loss in frequency domain, but at the cost of explosive primitive growth, resulting in two times of primitives compared to 3DGS. Although FDS-GS [50] proposes a frequency-aware density control method, they mainly focus on the connection between density and scale of primitives and do not really investigate how visual frequency impacts 3DGS. Compared to them, our method explicitly utilize the frequency information in training views as a prior to guide the densification, addressing the under-reconstruction and over-reconstruction in a simple and efficient scheme. Furthermore, we provide experimental evidence to prove that our method perpendicularly discovers under-reconstructed Gaussian primitives ignored by prior loss-driven method, which further emphasizes the superiority of our method.

## 3 Method

In this section, we introduce our CDC-GS in detail. In Section 3.1, we first review preliminaries about 3D Gaussian Splatting and Discrete Wavelet Transform. In Section 3.2, we introduce how CDC-GS computes the complexity and density of Gaussian primitives. In Section 3.3, we explain how CDC-GS utilizes complexity-density consistency to specify coarse and redundant primitives in under-reconstructed and over-reconstructed regions, which is the key of our method to allocate Gaussian primitives based on loss-agnostic visual prior.

### 3.1 Preliminary

#### 3.1.1 3D Gaussian Splatting

3D Gaussian Splatting (3DGS) represents a 3D scene using a set of Gaussian primitives $\{G_i\}_{i=1}^{N}$, where $N$ is the number of primitives. Each primitive $G_i$ is a 3D Gaussian distribution, parameterized

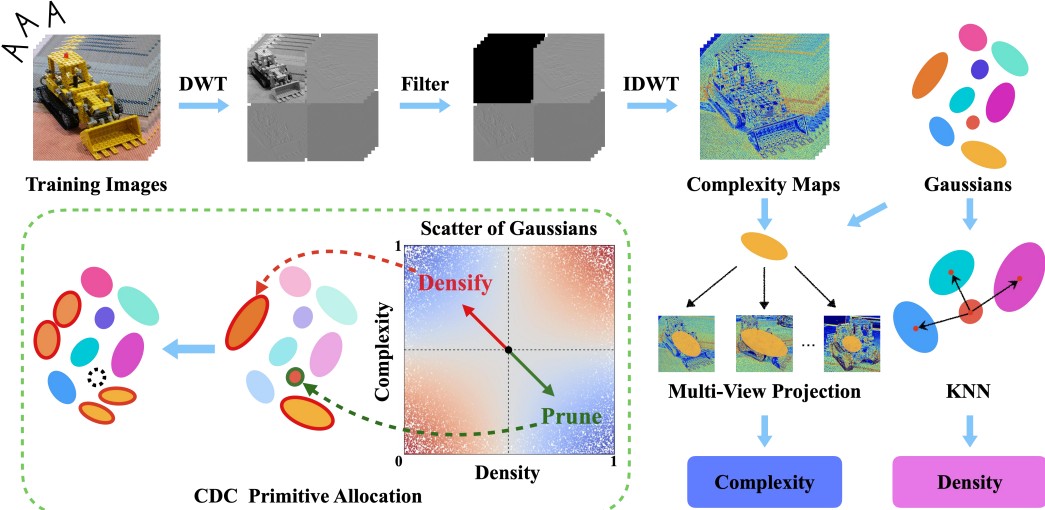

Figure 2: **Overview of our Complexity-Density Consistent Primitive Allocation.** We extract complexity maps for training images using DWT and back-projected the complexity maps onto Gaussian primitives to parameterize their complexity, while the density of each primitive is estimated with geometrical statistics about its neighbors. By densifying the primitives with low density vs. high complexity and pruning the primitives with high density vs. low complexity, CDC-GS adaptively allocates Gaussian primitives across the scene based on complexity of the space.

by its center $\mu_i \in \mathbb{R}^3$, opacity $o_i \in \mathbb{R}^1$, a full 3D covariance matrix $\Sigma_i \in \mathbb{R}^{3\times3}$, and Spherical Harmonic (SH) coefficients $c_i^{\mathrm{sh}} \in \mathbb{R}^{3\times16}$ encoding anisotropic appearance. The probability density function of the primitive is defined as,

$$G_i(x) = \exp\left(-\frac{1}{2}(x-\mu_i)^\top \Sigma_i^{-1}(x-\mu_i)\right),\tag{1}$$

where $x \in \mathbb{R}^3$ is an arbitrary point in 3D space. For rendering, primitives are projected to the image plane as 2D Gaussians $G_i^{\mathrm{2D}}$, and the color $C(x') \in \mathbb{R}^3$ of a pixel $x' \in \mathbb{R}^2$ is rasterized as,

$$C(x') = \sum_{j=1}^{P} c_j \alpha_j T_j, \quad \alpha_j = o_j G_j^{\mathrm{2D}}(x'), \quad T_j = \prod_{k=1}^{j-1}(1-\alpha_k),\tag{2}$$

where $P$ is the number of primitives covering pixel $x'$, and $c_j$ is the response of $c_j^{\mathrm{sh}}$ to the current camera view. The optimization of primitives is supervised with color rendering loss [1].

Besides the numerical optimization for primitives, the vanilla 3DGS adopts adaptive primitive allocation to express the scene better. Primitives receiving a positional gradient beyond threshold $\tau_{\mathrm{pos}}$ from the rendering loss are recognized as underfitting, and are densified to allocate more primitive to the region. Meanwhile, primitives possessing low opacity contribute little to modeling the scene according to Equation 2, which are pruned to reduce the primitive redundancy.

### 3.1.2 Discrete Wavelet Transform

2D Discrete Wavelet Transform (DWT) [9] is a potent tool to analyze frequency information of images. DWT decomposes an image into multiple sub-bands using a pair of orthogonal low-pass and high-pass filters, denoted by $F_{\mathrm{L}}$ and $F_{\mathrm{H}}$, respectively. Given a one-channel image $I \in \mathbb{R}^{H\times W}$, where H and W denote height and width of the image, one-level 2D Haar wavelet decomposition is formulated as,

$$\mathrm{DWT}(I) = \begin{bmatrix} I^{\mathrm{LL}} & I^{\mathrm{LH}} \\ I^{\mathrm{HL}} & I^{\mathrm{HH}} \end{bmatrix} = \begin{bmatrix} F_{\mathrm{L}} \\ F_{\mathrm{H}} \end{bmatrix} I \begin{bmatrix} F_{\mathrm{L}}^\top & F_{\mathrm{H}}^\top \end{bmatrix}.\tag{3}$$

Here, $I^{\mathrm{LL}}, I^{\mathrm{LH}}, I^{\mathrm{HL}}, I^{\mathrm{HH}}$ are four sub components of $I$ decomposed by DWT, where $I^{\mathrm{LL}}$ denotes the low-frequency structure (e.g., plain areas) and the rest three components correspond to high-frequency structure of the image (e.g., complex areas and edges). Moreover, DWT is an invertible transformation, which means one can obtain image $I$ with its four sub-components, as $I = \mathrm{IDWT}(I^{\mathrm{LL}}, I^{\mathrm{LH}}, I^{\mathrm{HL}}, I^{\mathrm{HH}})$.

## 3.2 Complexity and Density of Gaussian Primitives

Before introducing how our CDC-GS allocates Gaussian primitives with complexity-density consistency prior, we first define the complexity and density of each Gaussian primitive in this section.

The complexity of a primitive is to measure whether it's modeling a complex region, and we excavate this information from the frequency information of training views. We apply DWT to each training view $I$ and break it down into four sub-components, as defined in Equation 3. By removing the low frequency component $I^{\text{LL}}$ and retain the other three components, we extract the high frequency part of image $I$. And we define the complexity map of image $I$ as,

$$E_I = \text{IDWT}(0, I^{\text{LH}}, I^{\text{HL}}, I^{\text{HH}}), \tag{4}$$

which back projected to Gaussian primitives to compute the complexity of each primitive. Specifically, given a primitive $G_i$ involved to rendering pixel $x'$ with weight $w_i(x') = \alpha_i T_i$, the complexity of $G_i$ with respect to $x'$ is defined as $\gamma(G_i, x') = E_I(x') \cdot w_i(x') / \max_{j=1}^{P} (w_j(x'))$, where $P$ takes the same meaning as in Equation 2 and the max operator enumerates all the primitives covering $x'$ as normalization. Considering all $\gamma(G_i, \cdot)$, which is to enumerate all pixels covered by $G_i$ in all training views $\{I\}$, we take the maximum complexity to represent the complexity of $G_i$, which is formulated as,

$$\Gamma(G_i) = \max_{I_v \in \{I\}} \left( \max_{x' \in I_v} (\gamma(G_i, x')) \right). \tag{5}$$

The definition of $\gamma$ ensures that the complex areas in training views mainly affect the complexity of the primitives involved most in their rendering, and suppress their influence to the invisible primitives from the view.

Complexity alone cannot affirm if a primitive is in a under-reconstructed region or an over-reconstructed region. For example, high complexity region filled with dense Gaussian primitives is probably well reconstructed, while sparse primitives in such region typically indicate under-reconstruction, which can be ignored by loss-driven indicators such as gradient when the rendering loss is smoothed out with the high-frequency details. It's of necessity to cooperatively take both complexity and density of primitives to specify under-reconstruction and over-reconstruction. To estimate the density of the neighboring space of a primitive $G_i$, we compute the inverse of geometric mean of the center distance between $G_i$ and its neighbors, formulated as,

$$\Psi(G_i) = \left( \prod_{j \in \mathcal{N}_i} ||\mu_i - \mu_j|| \right)^{-\frac{1}{|\mathcal{N}_i|}}, \tag{6}$$

where $\mathcal{N}_i$ is the set of nearest neighbors of $G_i$. Both formulations are empirically validated to yield more stable and efficient primitive allocation (see Appendix B.1-B.2).

## 3.3 Complexity-Density Consistent Primitive Allocation

With the complexity and density of Gaussian primitive defined in Section 3.2, we can now introduce how the proposed CDC-GS allocates primitives based on them. As already discussed, the complexity and density of primitive are expected to share consistency in well-reconstructed region, otherwise it could indicate possibility of under-reconstruction or over-reconstruction. To formulate the consistency between the complexity and density of primitive $G_i$, we use the production of their normalized value,

$$s_i = \frac{\Gamma(G_i) - \Gamma_{\text{mean}}}{\Gamma_{\text{std}}} \cdot \frac{\Psi(G_i) - \Psi_{\text{mean}}}{\Psi_{\text{std}}}, \tag{7}$$

where $\Gamma_{\text{mean}}, \Psi_{\text{mean}}$ are the mean of complexity and density of all primitive respectively, and $\Gamma_{\text{std}}, \Psi_{\text{std}}$ denote the corresponding standard derivation. Positive value of $s_i$ indicates the consistency of complexity and density, while low complexity vs. high density and high complexity vs. low density results in negative $s_i$, indicating over-reconstruction and under-reconstruction respectively.

Our complexity-density consistency serves as a perpendicular complementary to existing primitive allocation method. Denoting primitives selected densified in a certain optimization step as $\mathcal{M}$, CDC-GS decomposes $\mathcal{M}$ into $\mathcal{M}_{\text{loss}}$ and $\mathcal{M}_{\text{cdc}}$, where $\mathcal{M} = \mathcal{M}_{\text{loss}} \cup \mathcal{M}_{\text{cdc}}$. $\mathcal{M}_{\text{loss}}$ refers to the primitives selected by existing loss-driven densification method (e.g., positional gradient), and $\mathcal{M}_{\text{cdc}}$

Table 1: **Quantitative comparison on novel view synthesis.** Results are reproduced with the official implementation of baselines. **Bold** indicates the best results. Our method consistently achieves superior performance across datasets with a fewer or comparable number of Gaussian primitives.

| Method | MipNeRF 360[30] | | | | Tanks & Temples[51] | | | | Deep Blending[52] | | | |
|---|---|---|---|---|---|---|---|---|---|---|---|---|
| | PSNR ↑ | SSIM ↑ | LPIPS ↓ | #GS ↓ | PSNR ↑ | SSIM ↑ | LPIPS ↓ | #GS ↓ | PSNR ↑ | SSIM ↑ | LPIPS ↓ | #GS ↓ |
| 3DGS[1] | 27.79 | 0.826 | 0.201 | 2.59M | 23.79 | 0.853 | 0.170 | **1.57M** | 29.73 | 0.906 | 0.238 | 2.47M |
| Taming-3DGS[8] | 27.91 | 0.821 | 0.211 | **2.53M** | 24.32 | 0.862 | 0.157 | 1.60M | 29.73 | 0.908 | **0.234** | **2.07M** |
| Ours (0.01) | **28.02** | **0.836** | **0.183** | **2.53M** | **24.42** | **0.866** | **0.149** | 1.60M | **29.84** | **0.909** | **0.234** | **2.07M** |
| AbsGS[4] | 27.72 | 0.835 | 0.169 | 4.06M | 23.22 | 0.856 | 0.152 | **1.90M** | 29.28 | 0.903 | 0.233 | 3.13M |
| Mini-Splatting-D[3] | 27.78 | 0.841 | **0.163** | 4.61M | 23.31 | 0.855 | 0.141 | 4.27M | 29.93 | 0.907 | **0.210** | 4.63M |
| Pixel-GS[5] | 27.84 | 0.835 | 0.176 | 5.27M | 23.66 | 0.856 | 0.150 | 4.50M | 28.95 | 0.896 | 0.247 | 4.65M |
| Ours (0.02) | **28.11** | **0.842** | 0.164 | **3.95M** | **24.47** | **0.867** | **0.140** | 2.56M | **30.03** | **0.912** | 0.218 | **2.49M** |

denotes the primitives selected by the proposed complexity-density consistency. For the primitives in under-reconstructed regions, we randomly sample up to $\rho_d \cdot N$ of them for densification, where the sampling probability is determined with $|s_i|$. For the primitives in over-reconstructed regions, we randomly sample up to $\rho_p \cdot N$ of them to be pruned, where the sampling probability is determined with $o_i$. The original pruning strategy of 3DGS at every 100 steps is removed, which is alternated to pruning all primitives with opacity below 0.1 for every 3000 steps.

To further improve the quality of $\mathcal{M}_{\text{loss}}$, relieving more primitives from under-reconstruction, we propose to adjust the densification threshold of each Gaussian primitive based on its complexity. The complexity-aware densification threshold is defined as,

$$\tau(\Gamma) = \tau_{\text{low}} + (\tau_{\text{upp}} - \tau_{\text{low}}) \cdot [1 - \sigma(\lambda \cdot \Gamma)], \tag{8}$$

where $\tau_{\text{low}}$ and $\tau_{\text{upp}}$ are the lower and upper bound of the densification threshold, respectively. $\sigma$ refers to the Sigmoid function, and $\lambda$ is a hyper-parameter.

## 4 Experiments

### 4.1 Experimental Settings

**Dataset.** We evaluate our method on the real-world datasets widely adopted in novel view synthesis. Specifically, we follow the standard protocol in vanilla 3DGS [1], using 9 scenes from the MipNeRF-360 [30] dataset, the playroom and drjohnson scenes from Deep Blending [52] dataset, and the train and truck scenes from Tanks & Temples [51] dataset. The preprocess of training data also strictly follows vanilla 3DGS [1].

**Metrics.** We report quantitative results with three standard metrics: Peak Signal-to-Noise Ratio (PSNR), Structural Similarity Index Metrics (SSIM) [53] and Learned Perceptual Image Patch Similarity (LPIPS) [54]. All metrics are computed on test views using the original evaluation protocol. We report the average score across all scenes for each dataset. Each experiment is repeated three times with different seeds, and we report the average performance across runs to account for randomness.

**Baselines.** We compare our method with representative methods investigating Gaussian primitive allocation, including vanilla 3DGS[1], Taming-3DGS[8] (constrained to the same number of primitives to our $\rho_d = 0.01$ version), AbsGS[4] (with the densification threshold set to 0.0004, the dense version of [4]), Mini-Splatting[3] (densification-only version without pruning), and Pixel-GS[5]. All baseline results are reproduced using the official implementations and default configurations to ensure fairness of experiments.

**Implementation Details.** Our method is implemented on top of the official 3DGS codebase [1]. We extract complexity map of input views using one-level Haar DWT. And the density of Gaussian primitives is computed with 3 nearest neighbors, which is to say $|\mathcal{N}_i| = 3$ for all primitives. We set $\rho_d = 0.01$ and $\rho_p = 0.01$ by default. We also evaluate an additional setting by increasing the splitting ratio to $\rho_d = 0.02$. Our complexity-aware densification threshold is configured with $\tau_{\text{low}} = 1.5 \times 10^{-4}$, $\tau_{\text{upp}} = 2 \times 10^{-4}$ and $\lambda = 6$. Other hyper-parameters are consistent with the vanilla 3DGS [1]. All experiments are performed on a single NVIDIA RTX 3090 GPU.

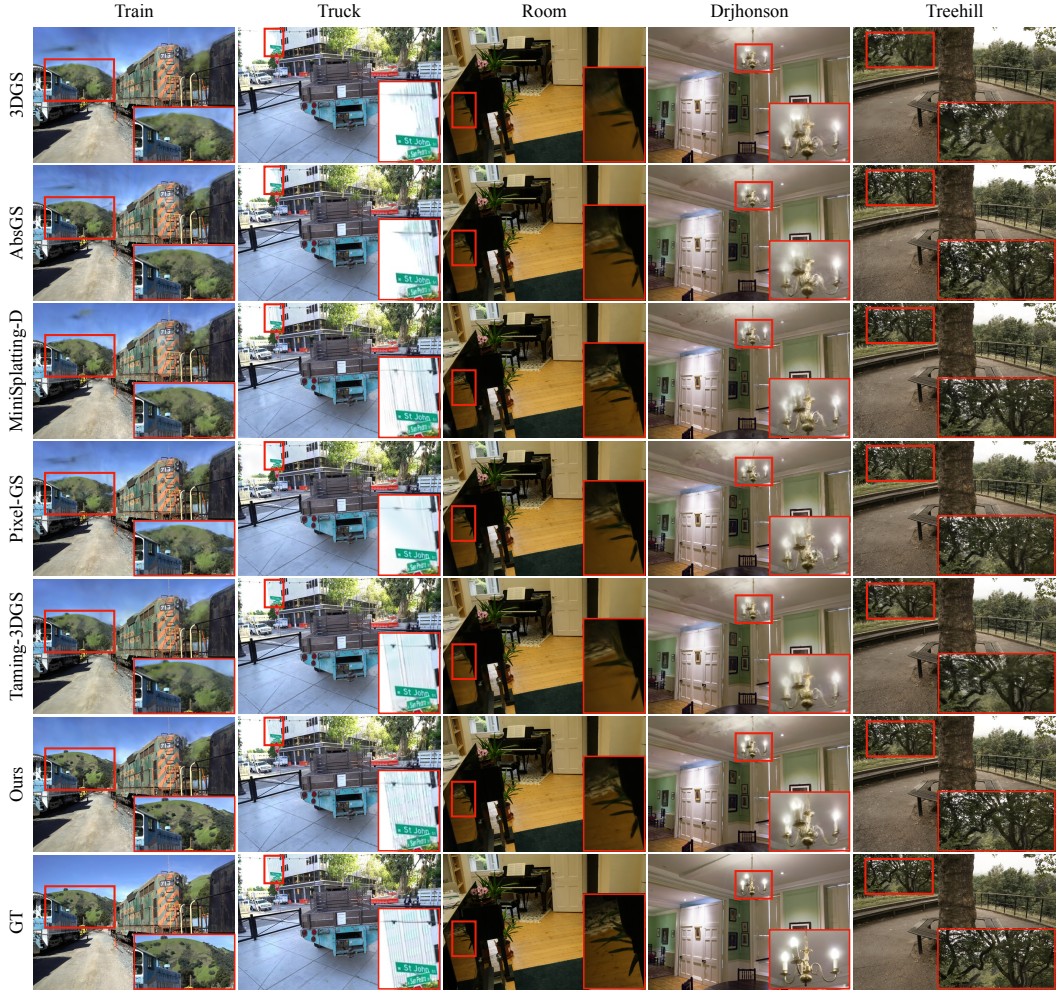

Figure 3: **Qualitative comparison on novel view synthesis results.** Compared to baseline methods, our method yields sharper textures, more faithful structures, and fewer artifacts without increasing primitives, which is attributed to our CDC strategy that improves the allocation of primitives.

## 4.2 Comparison to Baseline Methods

**Quantitative Comparison.** We report quantitative comparison between our CDC-GS and baseline methods in Table 1, where methods with approximate primitive number are grouped together for fair comparison. In the sparse group, CDC-GS ($\rho_d = 0.01$) consistently outperforms baselines, demonstrating superior fidelity and perceptual quality without increasing the number of primitives. In the dense group, we set $\rho_d = 0.02$ to match the primitive number of baselines, ensuring a fair and meaningful comparison. While utilizing fewer primitives in general, CDC-GS achieves the best rendering quality across all benchmarks. The results further validate that our proposed complexity-density consistent allocation strategy enables more precise modeling of intricate structures while improving primitive allocation efficiency.

**Qualitative Analysis.** We show qualitative comparisons corresponding to Table 1 across various scenes in Figure 3. Our method consistently presents more detailed textures, cleaner structural boundaries, and fewer artifacts. The results advocates that our method success to discover the under-reconstructed regions that the baseline methods tend to overlook.

## 4.3 CDC as a Perpendicular Densification Indicator to Rendering Loss

In this subsection, we reveal that the proposed complexity-density consistency serves as a perpendicular indicator against the loss-driven ones to specify the under-reconstructed regions.

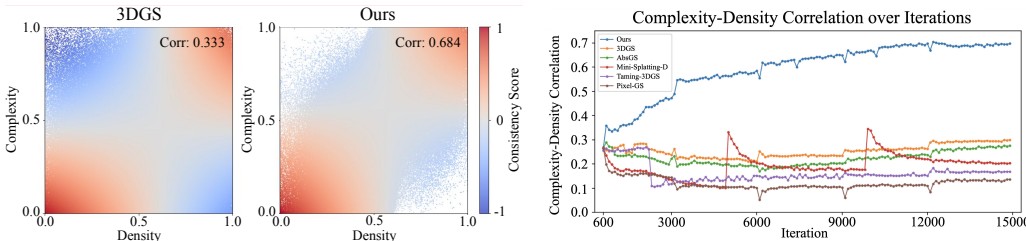

(a) Complexity-density map of primitives.  (b) Complexity-density correlation along densification.

Figure 4: **Analysis for complexity-density consistency on the scene "kitchen" of MipNeRF-360 dataset [30].** (a) The scatters depict the primitives at the end of densification (15k iteration) for 3DGS and our CDC-GS, with each primitive localized by its density and complexity. The color of each primitive indicates its consistency score $s_i$ (see Equation 7). The overall Pearson correlation [55] of complexity and density of primitives is shown at top-right. Color in the two scatters are normalized and aligned to ensure the same color corresponds to the same consistency score value. (b) Variation of complexity-density Pearson correlation over the densification progress (600–15k iteration).

In Figure 4a, we present two scatter diagrams for vanilla 3DGS [1] and CDC-GS, with all Gaussian primitives represented by their complexity and density. The scatters show the degree of the primitives in the optimized 3DGS model of scene "kitchen" matching the rule of complexity-density consistency. While vanilla 3DGS exhibits little consistency in complexity and density, CDC-GS significantly reduces the primitives in the left-up and right-down corner, which indicates the most inconsistent primitives are remove with our proposed primitive allocation strategy. Accompanied with Table 1 and Figure 3, we can draw the conclusion that the reduction of primitives in these regions do contribute to the quality of 3DGS model.

Moreover, in Figure 4b, we also provide the variation of complexity-density consistency along the training process of CDC-GS and all the baselines on the scene "kitchen". We use Pearson correlation [55] to evaluate the overall consistency between the complexity and density over all Gaussian primitives, which is to evaluate the degree of scatters in Figure 4 constrained in the first and the third quadrant. While CDC-GS performs a consistent increase in the consistency, baseline methods fail to largely improve the consistency alone the training process. Notably, slight improvement in the consistency of the baseline methods can be observed along the training process, which demonstrates that the complexity-density consistency naturally relates to the optimization of 3DGS, but struggles to improve without explicit manipulation. This advocates that the complexity-density consistency is perpendicular to loss-driven densification indicators.

To further consolidate this conclusion, we categorize the primitives in of vanilla 3DGS (left of Figure 4a) into two groups with the positional gradient of primitives and the densification threshold, which is presented in Figure 5a. We perform the same operation to AbsGS [4] and show the result in Figure 5b. The color of primitives indicates their positional gradient. Patterns can be barely drawn by splitting the primitives with the densification threshold, demonstrating that coarse primitives recognized with positional gradient ($\mathcal{M}_{\text{loss}}$, see Section 3.3) doesn't contain the primitives recognized with complexity-density consistency ($\mathcal{M}_{\text{cdc}}$), which is to say that $\mathcal{M}_{\text{cdc}}$ shares little cross with $\mathcal{M}_{\text{loss}}$. We believe the above experiment results are convincing enough to support the perpendicularity of complexity-density consistency against loss-driven densification indicators.

## 4.4 Ablations

We perform ablation studies on the Tanks & Temples dataset [51] using the vanilla 3DGS [1] as backbone to assess the individual contributions of the proposed Complexity-Aware Adaptive Threshold (CAAT) and Complexity-Density Consistency (CDC) introduced in Section3.3. Result is summarized in Table 2.

**Ablation on CAAT.** CAAT improves the reconstruction quality by manipulating densification threshold with complexity of Gaussian primitives. Slight increase in the number of primitives is observed because of aggressive densification in complex areas.

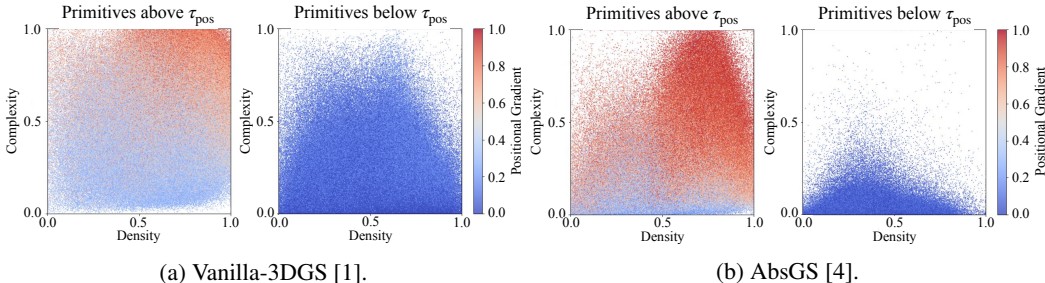

| | Primitives above $\tau_{pos}$ | Primitives below $\tau_{pos}$ | | Primitives above $\tau_{pos}$ | Primitives below $\tau_{pos}$ |

(a) Vanilla-3DGS [1].                    (b) AbsGS [4].

Figure 5: **Perpendicularity between complexity-density consistency and positional gradient.** We breakdown Figure 4a, categorizing the Gaussian primitives into two groups by if their positional gradient are above the densification threshold or not. And we show the same statistical information for AbsGS in Figure 5b. Color of primitives indicates the normalized magnitude of positional gradient. Primitives are categorized in a complexity-density consistency agnostic way, advocating that the proposed complexity-density consistency is perpendicular to the loss-driven positional gradient.

Table 2: **Ablation study on Tanks & Temples dataset.** We evaluate the contribution of the proposed modules: Complexity-Aware Adaptive Threshold (CAAT) and Complexity-Density Consistency (CDC). The full method (CDC-GS) combines both.

| Method | PSNR ↑ | SSIM ↑ | LPIPS ↓ | #GS ↓ |
|---|---|---|---|---|
| 3DGS | 23.79 | 0.853 | 0.170 | 1.57M |
| 3DGS + CAAT | 24.00 | 0.859 | 0.163 | 1.70M |
| 3DGS + CDC | 24.22 | 0.862 | 0.153 | **1.51M** |
| **CDC-GS (Ours)** | **24.42** | **0.866** | **0.149** | 1.60M |

**Ablation on CDC.** CDC achieves a substantial gain by simultaneous densification in under-reconstructed region and pruning in over-reconstructed region. Improvement in visual quality demonstrates that the proposed CDC strategy successfully mitigates the under-reconstructed problem in vanilla 3DGS[1], while the concurrent reduction of Gaussian primitives confirms its effectiveness in eliminating unnecessary primitives in low-complexity regions.

**Full Method.** When combining CAAT and CDC, the two modules act synergistically: CAAT introduces complexity-aware densification to refine local details, while CDC enforces global complexity-density consistency by adjusting the primitive distribution according to the complexity of the scene. As a result, CDC-GS significantly improve the reconstruction quality while maintaining a Gaussian primitive count comparable to the vanilla 3DGS[1], demonstrating the strength of the proposed Gaussian primitive allocation strategy.

## 5   Conclusion

In this paper, we propose CDC-GS, a novel Gaussian primitive allocation method, which is perpendicular to existing loss-driven ones, such as the ones using positional gradient. Our method explores the under-reconstructed region with high complexity and low density, and the over-reconstructed region with low complexity and high density. Extensive experiments are performed to demonstrate the superiority of our method and its perpendicularity to existing primitive allocation methods. It's power is fully played as a complementary to existing loss-driven primitive allocation methods. We believe it is valuable for finding the sub-optimally reconstructed regions ignored by existing methods in a prior driven manner.

**Limitations.** While improving Gaussian primitive allocation from an orthogonal perspective, our method still requires existing loss-driven cues (e.g., positional gradients), as the visual prior itself lacks direct awareness of reconstruction quality. In future work, we plan to further investigate how visual priors influence primitive allocation, toward more coherent and adaptive allocation strategies.

# Acknowledgments

The research was supported by the National Natural Science Foundation of China (U23B2009, 62471158).

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

## A  Impletation Details

### A.1  Algorithmic Implementation of CDC-GS

Algorithm 1 provides a detailed pseudocode of the training pipeline in CDC-GS. Built upon the vanilla 3DGS [1] framework, our procedure starts from a sparse initialization of Gaussian primitives and proceeds with joint optimization and CDC-guided density control across iterations. This pseudocode comprehensively implements the pipeline described in Section 3 of the main paper, integrating both loss-driven and prior-based cues for efficient and high-fidelity Gaussian allocation.

### A.2  Haar-based DWT Implementation

In our method, the visual complexity of each training view is extracted using a single-level 2D Discrete Wavelet Transform (DWT)[9]. We adopt the Haar wavelet basis due to its simplicity and efficiency. The filtering process is separable and implemented as two 1D convolutions along height and width, as defined in Equation 3 of the main paper, where $F_{\mathrm{L}} = [1, 1]/\sqrt{2}$, $F_{\mathrm{H}} = [-1, 1]/\sqrt{2}$ are the 1D low-pass and high-pass Haar filters. The resulting four subbands correspond to different frequency directions. Each 2D kernel derived from outer products of filters is summarized in Table 3.

Table 3: 2D Haar DWT convolution kernels derived from filter outer products.

| Subband | LL | LH | HL | HH |
|---|---|---|---|---|
| Kernel $K$ | $\frac{1}{2}\begin{bmatrix} 1 & 1 \\ 1 & 1 \end{bmatrix}$ | $\frac{1}{2}\begin{bmatrix} -1 & 1 \\ -1 & 1 \end{bmatrix}$ | $\frac{1}{2}\begin{bmatrix} -1 & -1 \\ 1 & 1 \end{bmatrix}$ | $\frac{1}{2}\begin{bmatrix} 1 & -1 \\ -1 & 1 \end{bmatrix}$ |

To compute the visual complexity of each image, we construct a high-pass filter by discarding the non-directional low-frequency subband $I^{\mathrm{LL}}$, and reconstruct the complexity map as defined in Equation 4 of the main paper. This effectively preserves only the high-frequency directional responses (vertical, horizontal and diagonal), enabling the resulting map $E$ to serve as a spatially-varying representation of complexity.

## B  More Ablations

### B.1  Comparison to Classical High-Pass Filters

To validate the effectiveness of our DWT-based design, we replace our DWT-based frequency estimation with three classical high-pass filters: Sobel[56], Scharr[57] and Laplacian[58], defined in Table 4, then integrate each into the CDC-GS framework without any other modifications.

Table 4: High-pass filter kernels used for comparison.

| Sobel | Scharr | Laplacian |
|---|---|---|
| $K_x = \begin{bmatrix} -1 & 0 & 1 \\ -2 & 0 & 2 \\ -1 & 0 & 1 \end{bmatrix}$ | $K_x = \begin{bmatrix} -3 & 0 & 3 \\ -10 & 0 & 10 \\ -3 & 0 & 3 \end{bmatrix}$ | |
| $K_y = \begin{bmatrix} -1 & -2 & -1 \\ 0 & 0 & 0 \\ 1 & 2 & 1 \end{bmatrix}$ | $K_y = \begin{bmatrix} -3 & -10 & -3 \\ 0 & 0 & 0 \\ 3 & 10 & 3 \end{bmatrix}$ | $K = \begin{bmatrix} 0 & -1 & 0 \\ -1 & 4 & -1 \\ 0 & -1 & 0 \end{bmatrix}$ |

As shown in Table 5, all methods achieve similar performance in rendering quality, demonstrating the robustness of our CDC-GS framework to the choice of complexity estimator. Notably, the DWT-based variant slightly outperforms the others in both rendering quality and Gaussian compactness. We

---

**Algorithm 1** CDC-GS Optimization and Density Control

---

**Input:** $w, h$: width and height of the training images

1: $M \leftarrow$ SfM Points          ▷ Positions
2: $S, C, A \leftarrow$ InitAttributes()      ▷ Covariances, Colors, Opacities
3: $i \leftarrow 0$          ▷ Iteration Count
4: $(I^{\text{LL}}, I^{\text{LH}}, I^{\text{HL}}, I^{\text{HH}}) \leftarrow$ DWT$(I)$      ▷ Eq.3
5: $E \leftarrow$ IDWT$(0, I^{\text{LH}}, I^{\text{HL}}, I^{\text{HH}})$      ▷ Eq.4
6: **while** not converged **do**
7:     $V, I \leftarrow$ SampleTrainingView()      ▷ Camera $V$ and Image $I$
8:     $\tilde{I} \leftarrow$ Rasterize$(M, S, C, A, V)$      ▷ Rendered Image $\tilde{I}$
9:     $L \leftarrow$ Loss$(\tilde{I}, I)$      ▷ Loss
10:     $M, S, C, A \leftarrow$ Adam$(\nabla L)$      ▷ Backprop & Step
11:     **if** IsRefinementIteration$(i)$ **then**
12:       $\Gamma \leftarrow$ Complexity$(M, S, C, A, E)$      ▷ Eq.5
13:       $\Psi \leftarrow$ Density$(M, S, C, A)$      ▷ Eq.6
14:       $s_i \leftarrow$ ConsistencyScore$(\Gamma, \Psi)$      ▷ Eq.7
15:       Compute sampling probabilities:
16:         $p_i^{\text{dens}} \propto |s_i|$ if $\Gamma_i > \mu_\Gamma, \Psi_i < \mu_\Psi$      ▷ High-complexity, Low-density
17:         $p_i^{\text{prune}} \propto \alpha_i$ if $\Gamma_i < \mu_\Gamma, \Psi_i > \mu_\Psi$      ▷ Low-complexity, High-density
18:       Sample $\mathcal{M}_{\text{cdc\_densify}} \sim$ Multinomial$(p^{\text{dens}}, \rho_d \cdot N)$
19:       Sample $\mathcal{M}_{\text{cdc\_prune}} \sim$ Multinomial$(p^{\text{prune}}, \rho_p \cdot N)$
20:       **for all** Gaussians $(\mu, \Sigma, c, \alpha)$ in $(M, S, C, A)$ **do**
21:         **if** $(\mu, \Sigma, c, \alpha) \in \mathcal{M}_{\text{cdc\_prune}}$ **then**
22:           RemoveGaussian$(\mu, \Sigma, c, \alpha)$      ▷ CDC Pruning
23:         **end if**
24:         **if** $(i \mod 3000 = 0)$ **and** $(\alpha < 0.1)$ **then**
25:           RemoveGaussian$(\mu, \Sigma, c, \alpha)$      ▷ Periodic Pruning
26:         **end if**
27:         $\tau(\Gamma) \leftarrow$ AdaptiveThreshold$(\Gamma)$      ▷ Eq.8
28:         $\mathcal{M}_{\text{loss}} \leftarrow \{G_i \mid \nabla_p L_i > \tau(\Gamma_i)\}$      ▷ Gradient-based
29:         $\mathcal{M}_{\text{densify}} \leftarrow \mathcal{M}_{\text{loss}} \cup \mathcal{M}_{\text{cdc\_densify}}$
30:         **if** $(\mu, \Sigma, c, \alpha) \in \mathcal{M}_{\text{densify}}$ **then**
31:           **if** $\|\Sigma\| > \tau_s$ **then**
32:             SplitGaussian$(\mu, \Sigma, c, \alpha)$      ▷ Over-reconstruction
33:           **else**
34:             CloneGaussian$(\mu, \Sigma, c, \alpha)$      ▷ Under-reconstruction
35:           **end if**
36:         **end if**
37:       **end for**
38:     **end if**
39:     $i \leftarrow i + 1$
40: **end while**
41: **return** Optimized Gaussian primitives $(M, S, C, A)$

---

attribute this to the multi-scale and multi-directional nature of the DWT, which enables more accurate modeling of visual complexity.

## B.2   Geometric vs. Arithmetic Mean

In Table 6, we compare two variants of our CDC-GS framework that differ only in how they compute the density of Gaussian primitives. Specifically, the Arithmetic Mean variant uses the inverse of the arithmetic mean of distances to neighboring Gaussians, whereas the Geometric Mean variant (our default, as defined in Equation 6 of the main paper) adopts the geometric mean formulation. Both variants are evaluated under identical settings.

Results show that using the geometric mean consistently yields slightly better rendering quality and reduces the number of Gaussians across all benchmarks. This confirms our choice, as the geometric

Table 5: **Comparison to classical high-pass filters on MipNeRF 360 [30].** We replace our DWT-based complexity estimation with Sobel, Scharr and Laplacian filters to construct complexity maps for CDC-GS. Our DWT-based formulation achieves the best performance across all metrics.

| Filter | PSNR ↑ | SSIM ↑ | LPIPS ↓ | #GS ↓ |
|---|---|---|---|---|
| Sobel | 27.96 | 0.834 | 0.186 | 2.60M |
| Scharr | 27.97 | 0.835 | 0.186 | 2.59M |
| Laplacian | 27.96 | 0.835 | 0.184 | 2.54M |
| DWT | **28.02** | **0.836** | **0.183** | **2.53M** |

mean provides a more sensitive estimation of local density, especially for short distances. We adopt the geometric mean formulation throughout the main paper.

Table 6: **Geometric vs. Arithmetic Mean in density computation.** We compare two variants of CDC-GS that differ only in whether the geometric or arithmetic mean is used for density estimation. The geometric mean consistently yields better performance with fewer Gaussians.

| Method | MipNeRF 360[30] | | | | Tanks & Temples[51] | | | | Deep Blending[52] | | | |
|---|---|---|---|---|---|---|---|---|---|---|---|---|
| | PSNR ↑ | SSIM ↑ | LPIPS ↓ | #GS ↓ | PSNR ↑ | SSIM ↑ | LPIPS ↓ | #GS ↓ | PSNR ↑ | SSIM ↑ | LPIPS ↓ | #GS ↓ |
| Arithmetic | 27.98 | 0.834 | 0.185 | 2.54M | 24.28 | 0.862 | 0.159 | 1.70M | 29.82 | **0.909** | **0.234** | 2.14M |
| Geometric | **28.02** | **0.836** | **0.183** | **2.53M** | **24.42** | **0.866** | **0.149** | **1.60M** | **29.83** | **0.909** | **0.234** | **2.07M** |

## B.3 Reverse-CDC Ablation

To further examine the role of complexity–density alignment, we designed a reverse variant in which Gaussians are densified in low-complexity & high-density regions and pruned in high-complexity & low-density ones, thereby inverting the CDC guidance.

Table 7: **Reverse-CDC ablation on MipNeRF 360 [30]**. Comparing vanilla 3DGS with Reverse-CDC shows degraded quality and increased number of Gaussians, highlighting the necessity of complexity–density alignment.

| Method | PSNR ↑ | SSIM ↑ | LPIPS ↓ | #GS ↓ |
|---|---|---|---|---|
| 3DGS | **27.79** | **0.826** | **0.201** | **2.59M** |
| Reverse-CDC | 27.62 | 0.817 | 0.224 | 2.71M |

As shown in Table 7, this reverse allocation leads to a degradation in rendering quality and a larger number of Gaussians. Although more primitives are introduced, the reconstruction becomes worse, demonstrating that enforcing complexity–density alignment is essential for efficient modeling.

## B.4 Efficiency Analysis of CDC-GS

We summarize the additional training cost introduced by the complexity and density modules. As shown in Table 8, the average overhead is dominated by the wavelet-based complexity computation, whereas both modules benefit from CUDA acceleration, keeping the cost practical. Although the training time increases, the rendering speed remains identical to vanilla 3DGS, and CDC-GS achieves higher reconstruction quality with a moderate additional cost.

# C  Additional Experiment Results

## C.1  Extended Visualization of Complexity-Density Correlation

As a supplement to the consistency analysis in Section 4.3 of the main paper, we provide additional visualizations of the complexity-density correlation variation on three scenes from different datasets: the outdoor scene "garden" from MipNeRF 360 [30], "train" from Tanks & Temples [51], and indoor

Table 8: **Training efficiency analysis of CDC-GS components on MipNeRF 360 [30].** "+ Density Only" and "+ Complexity Only" denote adding the KNN- and DWT-based modules individually.

| Method Variant | Training Time | Overhead |
|---|---|---|
| 3DGS (baseline) | 29:06 | - |
| + Density Only (KNN) | 30:40 | +1:34 |
| + Complexity Only (DWT) | 33:19 | +4:13 |
| CDC-GS (Full, Ours 0.01) | 35:36 | +6:30 |

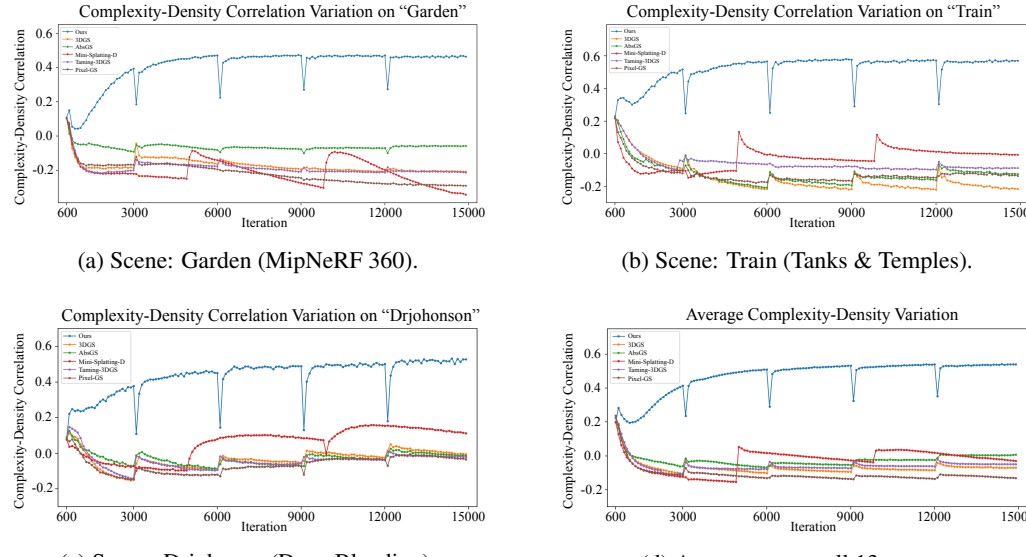

(a) Scene: Garden (MipNeRF 360).

(b) Scene: Train (Tanks & Temples).

(c) Scene: Drjohnson (Deep Blending).

(d) Average across all 13 scenes.

Figure 6: **Complexity-density correlation variation across datasets.** We show the Pearson correlation between visual complexity and local primitive density over training iterations. Our CDC-GS consistently improves and maintains this correlation, indicating stronger structural consistency compared to baseline methods.

scene "drjohnson" from Deep Blending [52]. We also report the average correlation variation across all 13 evaluated scenes.

These results in Figure 6 further validate the robustness of our CDC-GS framework in aligning visual complexity with local density across different scenes and datasets. In contrast, baseline methods often show unstable correlation trends or even negative correlation values, which contradict the expected relationship between structural complexity and spatial density, leading to persistent under-reconstruction in complex regions and over-reconstruction in simple areas.

## C.2 Supplementary Qualitative Results

As presented in Tables 9–12, our CDC-GS method consistently outperforms baselines across diverse benchmarks in terms of PSNR, SSIM, and LPIPS, while maintaining a significantly lower or comparable number of Gaussians. Specifically, under the same or fewer primitives, CDC-GS achieves better fidelity in high-complexity, detail-rich regions, thanks to its frequency-guided complexity modeling. This highlights the effectiveness of our complexity-density consistency strategy in allocating primitives to structurally complex areas, offering a complementary and orthogonal benefit to existing loss-driven methods. Our approach proves especially advantageous under constrained Gaussian budgets, delivering improved rendering quality without sacrificing efficiency.

Table 9: **Per-scene PSNR scores across three datasets. Bold** indicates the best results. Our method consistently achieves competitive or superior results across diverse scenes.

| Method | MipNeRF 360[30] | | | | | | | | | Tanks & Temples[51] | | Deep Blending[52] | |
| --- | --- | --- | --- | --- | --- | --- | --- | --- | --- | --- | --- | --- | --- |
| | Flowers | Treehill | Garden | Bicycle | Stump | Kitchen | Bonsai | Counter | Room | Train | Truck | Drjohnson | Playroom |
| 3DGS[1] | 21.92 | 22.86 | 27.88 | 25.68 | 26.91 | 31.42 | 32.47 | 29.18 | 31.81 | 22.07 | 25.51 | 29.37 | **30.09** |
| Taming-3DGS[8] | 21.81 | **23.08** | 27.78 | 25.47 | 26.63 | 32.05 | **32.84** | 29.38 | 32.12 | **22.62** | 26.01 | 29.43 | 30.03 |
| Ours (0.01) | 22.01 | 22.69 | 28.14 | 25.96 | 27.13 | 32.18 | 32.32 | 29.41 | 32.37 | 22.61 | 26.22 | 29.61 | 30.06 |
| AbsGS[4] | 21.78 | 22.19 | 27.87 | 25.76 | 27.04 | 31.48 | 32.32 | 29.18 | 31.87 | 21.14 | 25.30 | 28.61 | 29.96 |
| Mini-Splatting-D[3] | 21.89 | 22.59 | 27.80 | **26.00** | 27.46 | 31.48 | 32.35 | 28.62 | 31.83 | 21.22 | 25.39 | 29.40 | **30.47** |
| Pixel-GS[5] | 21.94 | 22.52 | 27.88 | 25.72 | 27.18 | 31.92 | **32.60** | 29.29 | 31.46 | 21.89 | 25.44 | 28.08 | 29.82 |
| Ours (0.02) | **22.14** | **22.74** | **28.26** | 25.99 | 27.39 | **32.25** | 32.38 | **29.51** | 32.33 | **22.74** | **26.19** | **29.68** | 30.39 |

Table 10: **Per-scene SSIM scores across three datasets. Bold** indicates the best results. Our method consistently achieves competitive or superior results across diverse scenes.

| Method | MipNeRF 360[30] | | | | | | | | | Tanks & Temples[51] | | Deep Blending[52] | |
| --- | --- | --- | --- | --- | --- | --- | --- | --- | --- | --- | --- | --- | --- |
| | Flowers | Treehill | Garden | Bicycle | Stump | Kitchen | Bonsai | Counter | Room | Train | Truck | Drjohnson | Playroom |
| 3DGS[1] | 0.622 | 0.652 | 0.875 | 0.779 | 0.783 | 0.933 | **0.948** | 0.916 | 0.929 | 0.820 | 0.885 | 0.905 | 0.907 |
| Taming-3DGS[8] | 0.613 | 0.646 | 0.871 | 0.775 | 0.771 | 0.930 | 0.946 | 0.910 | 0.922 | 0.831 | 0.893 | 0.908 | 0.908 |
| Ours (0.01) | **0.648** | **0.656** | **0.883** | **0.801** | **0.794** | **0.936** | 0.947 | **0.921** | **0.936** | **0.837** | **0.895** | **0.908** | **0.911** |
| AbsGS[4] | 0.654 | 0.645 | 0.883 | 0.801 | 0.795 | 0.935 | 0.951 | 0.920 | 0.935 | 0.822 | 0.890 | 0.899 | 0.907 |
| Mini-Splatting-D[3] | 0.659 | 0.659 | 0.884 | **0.811** | **0.816** | 0.936 | **0.953** | 0.918 | 0.936 | 0.819 | 0.890 | 0.906 | 0.908 |
| Pixel-GS[5] | 0.653 | 0.652 | 0.879 | 0.792 | 0.798 | 0.936 | 0.952 | 0.922 | 0.930 | 0.825 | 0.887 | 0.887 | 0.905 |
| Ours (0.02) | **0.660** | **0.660** | **0.887** | 0.808 | 0.812 | **0.938** | 0.947 | **0.925** | **0.940** | **0.838** | **0.895** | **0.912** | **0.912** |

Table 11: **Per-scene LPIPS scores across three datasets. Bold** indicates the best results. Our method consistently achieves competitive or superior results across diverse scenes.

| Method | MipNeRF 360[30] | | | | | | | | | Tanks & Temples[51] | | Deep Blending[52] | |
| --- | --- | --- | --- | --- | --- | --- | --- | --- | --- | --- | --- | --- | --- |
| | Flowers | Treehill | Garden | Bicycle | Stump | Kitchen | Bonsai | Counter | Room | Train | Truck | Drjohnson | Playroom |
| 3DGS[1] | 0.328 | 0.316 | 0.102 | 0.203 | 0.209 | 0.113 | 0.173 | 0.178 | 0.191 | 0.197 | 0.142 | 0.236 | 0.240 |
| Taming-3DGS[8] | 0.334 | 0.310 | 0.104 | 0.200 | 0.212 | 0.126 | 0.196 | 0.200 | 0.217 | 0.190 | 0.123 | 0.233 | **0.235** |
| Ours (0.01) | **0.274** | **0.289** | **0.090** | **0.175** | **0.206** | **0.109** | **0.165** | **0.167** | **0.170** | **0.181** | **0.118** | **0.232** | **0.235** |
| AbsGS[4] | 0.244 | **0.256** | 0.089 | 0.157 | 0.182 | 0.106 | 0.155 | 0.162 | 0.168 | 0.182 | 0.122 | 0.236 | 0.229 |
| Mini-Splatting-D[3] | 0.246 | **0.256** | 0.086 | **0.151** | **0.163** | 0.105 | **0.142** | **0.152** | **0.162** | 0.181 | **0.100** | **0.218** | **0.203** |
| Pixel-GS[5] | 0.252 | 0.270 | 0.093 | 0.173 | 0.180 | 0.106 | 0.161 | 0.162 | 0.183 | 0.179 | 0.120 | 0.255 | 0.240 |
| Ours (0.02) | **0.240** | **0.256** | **0.085** | 0.154 | 0.169 | **0.104** | 0.155 | 0.153 | **0.162** | **0.171** | 0.108 | 0.220 | 0.216 |

Table 12: **Per-scene number of Gaussian primitives across three datasets.** Values are reported in millions; **bold** indicates the lowest per scene. Our CDC-GS achieves competitive or significantly lower primitive counts, demonstrating its superior efficiency.

| Method | MipNeRF 360[30] | | | | | | | | | Tanks & Temples[51] | | Deep Blending[52] | |
| --- | --- | --- | --- | --- | --- | --- | --- | --- | --- | --- | --- | --- | --- |
| | Flowers | Treehill | Garden | Bicycle | Stump | Kitchen | Bonsai | Counter | Room | Train | Truck | Drjohnson | Playroom |
| 3DGS[1] | **2.75M** | **3.16M** | 3.65M | 4.68M | 4.14M | 1.54M | 1.07M | 1.04M | 1.24M | **1.09M** | **2.05M** | 3.11M | 1.84M |
| Taming-3DGS[8] | 3.09M | 3.90M | **3.62M** | **4.42M** | **3.92M** | 0.94M | 1.00M | 0.82M | 1.07M | 1.13M | 2.07M | **2.53M** | **1.60M** |
| Ours (0.01) | 3.09M | 3.90M | **3.62M** | **4.42M** | **3.92M** | 0.94M | 1.00M | 0.82M | 1.07M | 1.13M | 2.07M | **2.53M** | **1.60M** |
| AbsGS[4] | 5.16M | 6.13M | **4.68M** | 7.63M | 5.62M | 2.34M | **1.42M** | 1.49M | 2.12M | **1.63M** | **2.16M** | 4.04M | 2.22M |
| Mini-Splatting-D[3] | **4.76M** | **4.83M** | 5.44M | **5.85M** | **5.30M** | 3.69M | 3.76M | 3.83M | 4.06M | 3.95M | 4.58M | 4.91M | 4.35M |
| Pixel-GS[5] | 7.10M | 7.47M | 7.64M | 8.59M | 6.49M | 3.05M | 2.06M | 2.50M | 2.49M | 3.80M | 5.21M | 5.55M | 3.74M |
| Ours (0.02) | 5.14M | 5.39M | 5.67M | 7.05M | 6.21M | **1.50M** | 1.65M | **1.33M** | **1.64M** | 1.78M | 3.34M | **3.09M** | **1.89M** |

