# OpenReview forum: "Reframing Gaussian Splatting Densification with Complexity-Density Consistency of Primitives"
_NeurIPS.cc/2025/Conference — NeurIPS 2025 poster_

### Official Review · Reviewer_Jyxw · 2025-06-23

**Clarity:** 1
**Significance:** 3
**Originality:** 2
**Rating:** 3
**Confidence:** 4

**Summary:**

The authors propose a novel method for densifying and pruning Gaussian primitives with respect to the complexity of the training views and the density of the scene. The complexity is modeled with a high pass filter (2D DWT) and assigned to each Gaussian primitive via alpha compositing and the density is modeled with the average distance of each Gaussian to its three closest neighbors. Using both criteria for determining the densification strategy shows to improve the rendering quality without creating too many Gaussian primitives. The authors provide limited ablations and compare their results quantitatively to some of the current 3DGS methods.

**Questions:**

While the results show that the proposed method improves the rendering quality, the paper lacks in clarity and evaluation. Although the densification process is the main component of this work, it is not clear how primitives in under-/over-reconstructed regions are selected and how pd is determined. Further, the paper misses intuitive explanations, for instance: how formula 8 is designed, why the positional gradient is still required, or why the DWT is the best choice (other than it works best). Lastly, the paper introduces a lot of parameters while also changing some other components (such as the pruning interval), without proper evaluation / ablation. This makes it difficult to understand what component and which parameters have a big influence on the result.

Without further evaluation, comparisons and more clear explanations, I cannot confidently accept this paper as a significant contribution to the 3DGS research field.

**Ethical Concerns:**

["NO or VERY MINOR ethics concerns only"]

**Final Justification:**

I have raised my rating from reject to borderline reject. Although the authors have addressed all of my raised questions and the results outperform current 3DGS variants, I still believe that the paper lacks in originality, clarity and presentation. The contributions are rather small changes and it is not entirely clear which of the proposed components should be applied for future work (some promote compaction and some improve the visual quality). Nevertheless, I believe that with more fine grained evaluation and sensitivity measures, the paper has real potential.

**Limitations:**

yes

**Quality:**

2

**Strengths And Weaknesses:**

Strengths

- The topic is relevant
- The idea is interesting
- The results are promising
- The evaluation with different high pass filters underlines the choice for the Discrete Wavelet Transform (should be moved to the main paper)
- Figure 4a demonstrates well the effect of the methods

Warknesses:

In general, some parts are not clearly explained or confusing:
- Line 183: “Γmean, Ψmean are the mean of consistency and density” (should be complexity?)
- Line 195: “the sampling probability is determined with oi” what is oi and what is the intuition?
- Line 192: how is pd determined from |si| and what is the intuition? High values for |si| indicate good consistency. Why should those regions be further modified?
- Line 192: It is not clear how under-reconstructed regions and over-reconstructed regions are determined.
- Why is it required to still use the positional gradient?
- τlow and τup are set very similarly. It would be interesting if the quality already improves using a fixed threshold of τlow or τup.
- Where are all hyperparameters coming from? An ablation study would help.
- Line 230: What is ps?
- Why is Figure 4b looking completely different to the same graph shown in the appendix Figure 1?
- Formula 6: Why is the distance to the neighbors calculated with a product and not a sum?
- The advantages / reasons why the Discrete Wavelet Transform works best, are not discussed
- The pruning operation is changed without evaluation
- The runtime is likely increased due to the KNN calculations, but not mentioned in the paper
- The resolution of the rendered images in Figure 3 is too low
- The ablation study on T&T is not very expressive as it only includes two scenes
- More comparisons to related compaction methods from  https://arxiv.org/abs/2407.09510 should be included
- It would be interesting how well the method integrates into current SOTA methods

---

> ### Author Rebuttal · Authors · 2025-07-31
>
> **[Weaknesses 1]**
>
> Thank you for pointing out. The correct description should be: “Γmean, Ψmean are the mean of **complexity** and density”. We will fix it in the final version.
>
>
>
> **[Weaknesses 2, 3, 4]**
>
> We clarify the key concepts:
>
> 1. **Identifying Under-/Over-Reconstructed Regions**
>
>    As formalized in Equation 7 and Section 3.3 (Line 184), we use the **CDC score** *sᵢ* to measure local mismatch:
>
>    - *sᵢ* < 0 with **high complexity and low density** → **Under-reconstructed region** → Apply **densification**
>    - *sᵢ* < 0 with **low complexity and high density** → **Over-reconstructed region** → Apply **pruning**
>
>
> 2. **Pruning Sampling Based on Opacity *oᵢ***
>
>    The parameter *oᵢ* refers to the opacity of the Gaussian primitive, as introduced in Section 3.1.1 (Line 127). We use oᵢ as the sampling probability in pruning because opacity is a reliable indicator of a Gaussian’s contribution to the final rendering. Pruning requires more caution than densification, as mistakenly removing an important Gaussian can lead to unrecoverable degradation, while opacity oᵢ helps retain perceptually important Gaussians and avoids overly aggressive pruning.
>
> 3. **How $\rho_d$ Is Determined**
>
>    $\rho_d$ is a **fixed hyperparameter** (e.g., 0.01) controlling the proportion of Gaussians selected for densification. It is **not derived from the CDC score** *sᵢ*. Instead, we identify Gaussians with *sᵢ* < 0 (indicating a strong **mismatch** between complexity and density), and use multinomial sampling with probabilities proportional to |*sᵢ*| to prioritize the most structurally inconsistent Gaussians. Therefore, a high value of |*sᵢ*| does not indicate good consistency, but rather a stronger **inconsistency** — and thus a higher priority for refinement.
>
>
>
> **[Weaknesses 5, Questions: why the positional gradient is still required]**
>
> As noted in Section 4.5 (Line 293), we retain the positional gradient because our CDC prior improves allocation, but not per-view geometry. The gradient, derived from rendering loss, is essential for view-dependent refinement. We will clarify this dual-role design in the final version.
>
>
> **[Weaknesses 6, Questions: the paper misses intuitive explanations, for instance: how formula 8 is designed]**
>
> In Equation 8, $τ_{low}$ and $τ_{upp}$ are set very similarly because **densification in 3D Gaussian Splatting is highly sensitive** — even small threshold changes can cause large shifts in Gaussian count and quality. Our CAAT strategy adaptively modulates it between $τ_{low}$ and $τ_{upp}$ based on complexity and targets **only structurally complex regions for more aggressive densification while avoiding redundancy in simple areas**.
>
> **[Weaknesses 7, Questions: hyperparameters ablation.]**
>
> We provide ablations for all major hyperparameters on MipNeRF 360 dataset:
>
> - **KNN neighbor count |$N_i$|**. As shown in the table below, fewer neighbors (e.g.,  |$N_i$| = 1, 2) improves quality slightly by enhancing sensitivity to local structures, but also increases noise and instability. Therefore, we select |$N_i$| = 3 as a balanced choice.
>
>   | \|$N_i$\| | PSNR ↑ | SSIM ↑ | LPIPS ↓ | #GS ↓ |
>   | --------- | ------ | ------ | ------- | ----- |
>   | 1 | 28.05| 0.838| 0.179 | 2.51M |
>   | 2 | 28.03| 0.837 | 0.179 | 2.54M |
>   | 3 | 28.02| 0.836 | 0.183 | 2.53M |
>   | 4 | 28.02| 0.836 | 0.182 | 2.55M |
>   | 5 | 28.01| 0.836 | 0.183 | 2.55M |
>
> - **$τ_{low}$, $τ_{upp}$ (CAAT Module)**. Ablation shows that reducing the fixed threshold from 0.0002 to 0.0001 improves PSNR (+0.53 dB) but nearly triples gaussian count (2.74M→7.89M). This highlights the sensitivity of the threshold and motivates our adaptive design. We set $τ_{upp}$ = 0.0002 and $τ_{low}$ = 0.00015 to improve quality with minimal overhead. Due to characters limitation, we omit the full table here but will include it in the final version.
>
> - **λ (CAAT Module)**. Ablation varying λ from 1 to 10 shows that **the method is not sensitive to the value of λ**. PSNR varies within 0.07 dB and gaussian count increases slightly and gradually, with no sharp drop in performance. The default setting λ = 6 achieves a balanced trade-off across scenes. Due to rebuttal length limits, we omit full tables but will include them in the final version.
>
>
>
> **[Weaknesses 8]**
>
> This is a typographical error, and the correct symbol should be $\rho_d$ instead of $\rho_s$ in Line 230. We will correct it in the final version.
>
>
>
> **[Weaknesses 9]**
>
> Thank you for pointing this out. The figures reflect real training behaviors on different scenes and are consistent with our method.
>
> - **Scene differences**: Figure 4b shows *kitchen* (MipNeRF 360), while Appendix Fig.1 includes *garden*, *train*, *drjohnson*, and averages across 13 scenes. Differences in geometry and frequency naturally lead to trend variation, but the core finding remains: **CDC-GS consistently improves complexity-density correlation**, unlike baselines.
> - **Multi-factor influence**: While our CDC-GS promotes complexity-density alignment, downstream optimization can cause scene-specific fluctuations. Nevertheless, **an overall upward trend is observed across scenes**, validating the method’s robustness.
> - **Periodic pruning**: The drops every 3000 iterations are due to pruning low-opacity Gaussians (<0.1), which momentarily alters density. The rapid recovery confirms that **our method is adaptive and resilient to structural changes**.
>
>
>
> **[Weaknesses 10]**
>
> We use the geometric mean for density estimation (Eq. 6) as it better captures local crowding — one close neighbor strongly lowers the mean. As shown in **Supplementary B.2**, this leads to better quality and fewer Gaussians than the arithmetic mean.
>
>
>
> **[Weaknesses 11, Questions: why the DWT is the best choice]**
>
> We chose DWT for the following key reasons:
>
> - **Multi-scale decomposition**: Unlike single-scale filters, DWT captures both fine and coarse structural details, crucial for modeling spatially varying 3D complexity.
> - **Directional sensitivity**: DWT provides high-frequency responses in multiple orientations (LH, HL, HH), enabling richer structural analysis than edge-only filters.
> - **Frequency separation**: By discarding the LL band and inverting DWT, we isolate high-frequency components, preserving more relevant geometric cues.
> - **Empirical support**: DWT-based approach consistently improves rendering quality and reduces Gaussian count across scenes.
>
>
>
> **[Weaknesses 12]**
>
> We now include the missing pruning ablation results on MipNeRF 360:
>
> |**Method**|**PSNR ↑**|**SSIM ↑**|**LPIPS ↓**|**#GS ↓**|
> | --------------------- | ---------- | ---------- | ----------- | --------- |
> |3DGS|27.79|0.826|**0.201**| 2.59M|
> |3DGS + periodic prune|**27.81**|**0.828**|0.205|**1.79M**|
>
> This demonstrates that our pruning strategy significantly reduces the number of Gaussians without degrading quality.
>
> The motivation is, overlapping Gaussians in dense regions often have low opacity due to alpha-blending. These are likely redundant and can be safely removed. Periodic pruning thus improves compactness and provides a more accurate density estimate for CDC-based allocation.
>
>
>
> **[Weaknesses 13]**
>
> We conducted an ablation study on the **MipNeRF 360** dataset to quantify the additional training cost of our complexity and density components:
>
> | **Method Variant**| Training Time (overhead) |
> | :---------------------- | :----------------------- |
> | 3DGS (baseline) | 29:06 |
> | + Density Only (KNN)| 30.40 (+1:34) |
> | + Complexity Only (DWT) | 33.19 (+4:13) |
> | Full (Ours 0.01) | 35.36 (+6:30) |
>
> **KNN over millions of Gaussians is expensive**, but we mitigate this via **CUDA-based acceleration** to fully leverage GPU parallelism, keeping the density overhead modest. Likewise, **complexity computation is integrated into the alpha-blending stage of 3DGS rendering to reduce cost.**
>
> Notably, **rendering speed remains unaffected**. Our method maintains the same FPS with vanilla 3DGS while improving reconstruction quality.
>
>
>
> **[Weaknesses 14]**
>
> Figure 3 uses reduced resolution to meet NeurIPS file limits. We will replace it with a higher-quality version in the final paper.
>
>
>
> **[Weaknesses 15]**
>
> We extended the ablation to all 9 MipNeRF 360 scenes. Results match main paper conclusions:
>
> |Method|PSNR ↑|SSIM ↑| LPIPS ↓ | # Gaussians |
> | ------------- | --------- | --------- | --------- | ----------- |
> | 3DGS| 27.79| 0.826| 0.201| 2.59M |
> | + CAAT| 27.86| 0.829| 0.197| 2.82M |
> | + CDC | 27.92| 0.832| 0.189| 2.40M |
> | Full (CDC-GS) | **27.98** | **0.834** | **0.185** | **2.53M**|
>
>
>
> **[Weaknesses 16]**
>
> We include comparisons with representative compaction methods (Mini-Splatting, LightGaussian, Taming-3DGS) proposed in the paper suggested by the reviewer, and will incorporate related references accordingly in the final version:
>
> | **Method** | **PSNR ↑** | **SSIM ↑** | **LPIPS ↓** | **#GS ↓** |
> | --------------- | ---------- | ---------- | ----------- | --------- |
> | 3DGS | 27.79 | 0.826 | **0.201**| 2.59M|
> | Mini-Splatting| 27.35 | 0.822 | 0.218 | **0.49M** |
> | LightGaussian| 27.27| 0.804 | 0.241| 0.81M|
> | Taming-3DGS| 27.80 | 0.815 | 0.225| 1.47M|
> | Ours Prune Only | **27.82** | **0.825** | 0.214 |1.47M|
>
> ​The results show that Our pruning-only variant not only **reduces the number of Gaussians by over 1.1 million**, but also **slightly improves rendering quality** compared to vanilla 3DGS.
>
>
>
> **[Weaknesses 17]**
>
> We integrated CDC into 3DGS-MCMC. With the same number of Gaussians, performance improves:
>
> | **Method** | **PSNR ↑** | **SSIM ↑** | **LPIPS ↓** | **#GS ↓** |
> | --------------- | ---------- | ---------- | ----------- | --------- |
> | Ours (0.01)|28.02|0.836| 0.183| 2.53M|
> | 3DGS-MCMC| 28.13 |0.842| 0.179 | 2.53M|
> | 3DGS-MCMC + CDC |**28.24**|**0.845**| **0.175** | 2.53M|
>
> This confirms that CDC is **plug-and-play** and can enhance existing 3DGS variants.
>
>
> ​We will include these analyses in the final version and sincerely appreciate the reviewer’s constructive feedback.

---

> > ### Comment · Reviewer_Jyxw · 2025-08-01
> >
> > Dear Authors,
> >
> > Thank you for clarifying a lot of things! The additional comparisons and explanations really help the understanding. The following three questions still remain:
> >
> > ---
> > Q1. It is still not clear to me which configuration is supposed to be the final proposed method of this paper. In Table 1 in the paper, there are already two versions with (pd=0.01) and (pd=0.02). And now with the additional experiments there are new comparisons with “Ours Prune Only”. Is that configuration without CAAT? Additionally, when adding MCMC, CAAT is not applied. This makes it difficult to understand which component to apply in which scenarios.
> >
> >
> > |Method	                       | PSNR ↑	|SSIM ↑	|LPIPS ↓	|#GS ↓|
> > |---|---|---|---|---|
> > |3DGS-MCMC + CDC	|28.24	|0.845	|0.175	|2.53M|
> > |Ours (Prune Only)		|27.82	|0.825	|0.214	|1.47M|
> > |Ours (0.01)			|28.02 	|0.836 	|0.183 	|2.53M|
> > |Ours (0.02) 			|28.11 	|0.842 	|0.164 	|3.95M|
> >
> > While this is not a major weakness, an intuitive guideline on when to use which of the components would help the understanding. Especially when including the proposed methods in future work, it is not obvious which parts should be used to achieve which results.
> >
> > ---
> >
> > Q2. The ablation with “3DGS + periodic prune” seems to have a significant impact on the number of Gaussian primitives. This modification should be treated separately to CAAT and CDC in the ablation study and CDC and CAAT should be tested individually without using the pruning strategy. Otherwise it is for instance not clear if CAAT or CDC increase or decrease the number of Gaussian primitives.
> >
> > ---
> >
> > Q3. Regarding point 3: “Therefore, a high value of |sᵢ| does not indicate good consistency, but rather a stronger inconsistency — and thus a higher priority for refinement.”
> >
> > In my understanding negative s_i indicates inconsistency and positive s_i consistency. Why should the absolute value determine stronger inconsistency? A high absolute value of s_i could also mean that the consistency is very high. Or am I missing something?

---

> ### Author Response · Authors · 2025-08-03
>
> **[Questions 1]**
>
> Thank you for raising this important point. To clarify:
>
> - Our **default proposed method** is "**Ours (0.01)**", which includes **CDC (with both densification and pruning) + CAAT + periodic pruning**, as shown in the main paper (Table 1).
> - "**Ours (0.02)**" is a variant with a **more aggressive densification** budget ($\rho_d$ = 0.02), leading to higher quality but more gaussians for fair comparison with other 3DGS baseline methods that use **a similarly high number of gaussians**.
> - "**Ours Prune Only**" is an ablation variant that **removes CAAT and densification in CDC**, keeping **periodic pruning and CDC pruning** to test its standalone effect.
> - When integrating into **3DGS-MCMC**, we **only use CDC (with both densification and pruning)**, since 3DGS-MCMC already controls primitive count, making CAAT and pruning less relevant.
>
> We will clearly annotate each configuration and its components in the final version. For practical usage:
>
> | **Scenario**         | **Components to Apply**                        | **Use Case**                     |
> | -------------------- | ---------------------------------------------- | -------------------------------- |
> | Standard 3DGS        | CDC ($\rho_d$=0.01) + CAAT + periodic pruning        | Balanced quality and compactness |
> | High-fidelity need   | CDC ($\rho_d$=0.02) + CAAT + periodic pruning        | Best quality but higher cost     |
> | Memory-limited setup | Prune Only (CDC pruning + periodic pruning)    | Quality-preserving compression   |
> | With 3DGS-MCMC       | CDC Only (with both densification and pruning) | Enhances MCMC’s allocation       |
>
> **[Questions 2]**
>
> We clarify that the **+CAAT results** in the rebuttal **did not use periodic pruning**, while the **+CDC results** **included** it, since we treat periodic pruning as an integral part of the CDC module. To isolate the impact of each component, we conducted an **additional ablation** on the MipNeRF 360 dataset, **removing periodic pruning from the +CDC setting**:
>
> | Ablations                       | PSNR ↑ | SSIM ↑ | LPIPS↓ | Gaussians↓ |
> | ------------------------------- | ------ | ------ | ------ | ---------- |
> | 3DGS                            | 27.79  | 0.826  | 0.201  | 2.59M      |
> | + CAAT (without periodic prune) | 27.86  | 0.829  | 0.197  | 2.82M      |
> | + CDC (without periodic prune)  | 27.92  | 0.831  | 0.187  | 3.29M      |
> | + CDC (with periodic prune)     | 27.92  | 0.832  | 0.189  | 2.40M      |
>
> These results indicate:
>
> - **CAAT** slightly increases the number of gaussians (+0.23M) while consistently enhancing quality (+0.07dB PSNR).
>
> - **CDC** brings a more significant quality boost (+0.13dB PSNR), but increases gaussians substantially (+0.7M).
>
> - **Periodic pruning** effectively reduces the number of gaussians (−0.89M) **without compromising rendering quality**, and serves as a good complement to CDC.
>
> We will include these results in the final version to clearly illustrate their effect on the number of Gaussian primitives. Thank you again for raising this important question.
>
> **[Questions 3]**
>
> You are absolutely right that a **negative sᵢ indicates inconsistency**, while **positive sᵢ indicates consistency**. We clarify that:
>
> - **We only perform CDC densify/prune on Gaussians where sᵢ < 0**, i.e., when the complexity and density are mismatched (e.g., high complexity but low density, or vice versa).
> - For these inconsistent Gaussians, we use **|sᵢ|** to measure the **degree of inconsistency** — the higher the absolute value, the more severe the mismatch.
> - **Gaussians with sᵢ > 0 are will not be refined** regardless of their |sᵢ| value, since they already align in structure.
>
> This ensures that our method focuses exclusively on **inconsistency correction**, not on suppressing highly consistent gaussians.
>
> We will clarify this in the final version to avoid misunderstanding. Thank you for this important clarification request.

---

> > ### Comment · Reviewer_Jyxw · 2025-08-05
> >
> > Dear Authors,
> > Thank you for your response!
> >
> > I have just compared your reported results to the results of the MCMC paper. They tend to differ quite a lot:
> >
> > Your result:
> > 3DGS-MCMC: 	28.13	0.842	0.179	2.53M
> >
> > Reported result in MCMC:
> > MCMC: 29.89 / 0.90 / 0.19
> >
> > Can you please clarify the difference?

---

> > > ### Author Response · Authors · 2025-08-05
> > >
> > > Thank you for this question.
> > >
> > > First, we clarify that the 3DGS-MCMC paper evaluates on only **7 out of the 9 scenes** in the **MipNeRF 360** dataset (excluding *flowers* and *treehill*), as explicitly mentioned in their paper.
> > > In contrast, we report 3DGS-MCMC results over **all 9 scenes** to align with the **Ours (0.01)** setting for fair comparison.
> > > This difference in scene coverage contributes to the numerical discrepancy in reported metrics.
> > >
> > > Second, the 3DGS-MCMC method includes a mechanism to **constrain the total number of Gaussians via a configurable upper bound**.
> > > To ensure a fair comparison, we set this upper bound to **match the final number of Gaussians used in our method** for each scene.
> > > As a result, the number of Gaussians in our reported results is **lower** than that used in original 3DGS-MCMC paper, which employed a higher budget.
> > > Since **rendering quality is closely related to the number of Gaussians**, this also explains the performance gap between our reproduced numbers and the original paper.
> > >
> > > We will clearly state these differences in the final version.

---

### Official Review · Reviewer_qsuh · 2025-06-23

**Clarity:** 3
**Significance:** 2
**Originality:** 3
**Rating:** 4
**Confidence:** 4

**Summary:**

This paper proposes a novel method for allocating primitives in 3D Gaussian Splatting. The authors introduce measures of complexity and density for Gaussian primitives to identify under-reconstructed and over-reconstructed regions, and develop an allocation strategy based on these metrics. Experimental results show that the proposed method outperforms baseline approaches overall.

**Questions:**

My main concern is that the paper does not fully discuss how uneven complexity-density distributions affect rendering performance. I do not doubt the relevance of these two metrics to the 3DGS model, but I believe a deeper discussion is necessary. If the authors could provide a more thorough justification or analysis of this point, I would consider raising my score.

**Ethical Concerns:**

["NO or VERY MINOR ethics concerns only"]

**Final Justification:**

The authors have answered my question well, and I have no further concerns. I have raised my score to 4. However, I still believe the work does not provide additional insights for related research, so I do not consider it deserving of a score of 5.

**Limitations:**

Yes, the author answered the questions in checklist well and the limitations are fully addressed.

**Paper Formatting Concerns:**

The paper is clearly and effectively written.

**Quality:**

3

**Strengths And Weaknesses:**

### Strengths:
- S1. The writing is clear and effectively presents main idea. The methodology is straightforward, and the experimental details are well-presented.

- S2. Most of the quantitative results outperform baselines.

### Weaknesses:
- W1: Figure 4 shows that the proposed method achieves a higher correlation between complexity and density compared to baseline methods. However, since the allocation strategy is explicitly designed based on these two metrics, it is expected that it would outperform baselines on this specific correlation measure. This raises concerns about the fairness of using this metric to claim the method in the paper is better. Moreover, in Figure 4b, the authors illustrate how the correlation score evolves during training. If a higher correlation is considered indicative of better scene modeling or rendering quality, then 3DGS should perform better than other baseline methods. However, the quantitative results do not consistently reflect this. This discrepancy suggests that the correlation metric may not directly align with final rendering performance, and the paper does not sufficiently discuss or justify this gap.

---

> ### Author Rebuttal · Authors · 2025-07-30
>
> **[Weaknesses, Questions]**
>
> ​We sincerely thank you for this thoughtful comment. We agree that using complexity-density correlation (CDC) as an evaluation metric requires careful justification, especially given that our method is explicitly designed based on these two factors. Below, we provide a more thorough explanation, and include additional experiments to demonstrate how uneven CDC distributions affect rendering outcomes.
>
> To answer the reviewer's concern - "**3DGS should perform better than other baseline methods. However, the quantitative results do not consistently reflect this**":
>
> - **CDC is not the only factor that affects rendering performance**. In our method, CDC primarily serves as a **guiding prior** in the densification stage, aiming to better allocate Gaussian primitives in accordance with the scene’s structural complexity. The correlation metric reflects how well the primitive distribution matches the structural needs of the scene, but **the final rendering quality also depends on subsequent stages** such as opacity-guided optimization and rasterization, as demonstrated in prior 3DGS works. Thus, **high CDC alone does not deterministically lead to better rendering, but rather provides a principled foundation for better primitive allocation**,  especially under a constrained primitive budget. It is not a deterministic predictor of final rendering score. We will revise the manuscript to clarify this distinction.
> - **Minor CDC gains may not yield visible improvements.** While vanilla 3DGS does show a slight CDC improvement at later stages (as noted in Fig. 4b), the Pearson coefficients remain low (typically 0.1–0.3), which we consider insufficient to meaningfully affect rendering. In contrast, our method achieves a much higher CDC (~0.7), which we believe leads to more accurate primitive allocation and ultimately contributes to improved rendering under similar budgets. Thus, the reviewer’s concern highlights an important nuance: **CDC matters, but only when enforced strongly and consistently.**
> - **Other methods optimize rendering performance from different perspectives**. For example, baselines like Mini-Splatting and AbsGS enhance densification through techniques such as depth-based splitting and absolute gradient accumulation. **These strategies do not explicitly aim to align complexity and density**, so their rendering improvements may not correlate with CDC scores. This further justifies our choice to evaluate CDC-guided densification in isolation, rather than treating CDC as a universal indicator of rendering quality.  **Therefore, it is reasonable that quantitative rankings in rendering performance do not always align with CDC scores across all methods.**
> - **We further support this argument with a reverse ablation on MipNeRF 360 dataset.** To assess the impact of CDC more directly, we design a control variant called **3DGS + reverse CDC**, where we intentionally densify Gaussians in low-complexity & high-density regions, and prune those in high-complexity & low-density ones — essentially flipping the CDC guidance.
>
>   | **Method**         | **PSNR ↑** | **SSIM ↑** | **LPIPS ↓** | **#GS ↓** |
>   | ------------------ | ---------- | ---------- | ----------- | --------- |
>   | 3DGS (baseline)    | **27.79**  | **0.826**  | **0.201**   | **2.59M** |
>   | 3DGS + reverse CDC | 27.62      | 0.817      | 0.224       | 2.71M     |
>
>   As shown in above, the **reverse CDC variant degrades rendering performance** (PSNR −0.17 dB, SSIM −0.009, LPIPS +0.023), and also increasing the number of Gaussians, indicating inefficient allocation. This result empirically confirms that **disrupting the complexity-density alignment harms rendering qaulity, while reinforcing it (as our method does) leads to improvement.**
>
> - **Theoretically, CDC fits the nature of Gaussian splatting.** Since Gaussians are ellipsoidal and composited via alpha-blending, **a higher number of Gaussians in structurally complex regions enables better approximation of fine geometry and texture**. Conversely, smooth areas (e.g., walls) require fewer primitives. Aligning density with visual complexity avoids both under- and over-reconstruction, which we visualize in Figure 1.
>
> ​We will incorporate these clarifications in the final version, and we thank the reviewer again for highlighting this important aspect. We hope this additional analysis supports your consideration to raise the score.

---

### Official Review · Reviewer_MnE1 · 2025-06-30

**Clarity:** 4
**Significance:** 4
**Originality:** 3
**Rating:** 4
**Confidence:** 4

**Summary:**

This paper proposes a novel densification strategy that accounts for both the complexity and density of Gaussian primitives. To achieve this, the authors define the complexity of Gaussians using the discrete wavelet transform (DWT) and the density of Gaussians based on the distance between neighboring Gaussians. Leveraging these two concepts, they introduce a Complexity-Density Consistency (CDC) score to determine which Gaussians deserve densified and a Complexity-Aware Adaptive Threshold (CAAT) to adaptively prune Gaussians based on both gradient and complexity. Experimental results show that the proposed method outperforms the rendering quality of existing Gaussian densification approaches.

**Questions:**

- In the definition of complexity, why do the authors consider the maximum complexity value for all pixels in training views, instead of an average of them?

- It would be valuable to investigate whether the proposed densification strategy remains effective when targeting a smaller number of Gaussians. To this end, I recommend conducting additional experiments with a more aggressive pruning configuration aimed at reducing the number of Gaussians. These results are recommended to be compared with existing compact 3DGS baselines, such as LightGaussian [NeurIPS'24], Mini-Splatting (not Mini-Splatting-D) [ECCV'24], and Taiming-3DGS [SIGGRAPH Asia'24].

- As I mentioned above, additional costs for computing complexity and density might be required. Please report the comparison for the training time of each method.

- It would be better to add 3DGS-MCMC [NeurIPS'24] in comparison.

**Ethical Concerns:**

["NO or VERY MINOR ethics concerns only"]

**Final Justification:**

The authors have addressed all of my concerns during the rebuttal period. The additional evaluations strengthen the clarity of the proposed method. Notably, it is interesting that it achieves better performance when combined with 3DGS-MCMC, as well as under the aggressive pruning setup. Therefore, I will maintain my positive rating.

**Limitations:**

- Additional computational costs are required for computing the densification standard.
- Also, limited technical contribution might be seen as a lack of novelty.

**Paper Formatting Concerns:**

There are no paper formatting concerns.

**Quality:**

4

**Strengths And Weaknesses:**

### Strengths
- The authors address the limitation of the prior loss-based densification criterion and introduce two important factors for densification, complexity and density of Gaussians.
- It achieves superior rendering quality compared to existing Gaussian densification approaches, such as Mini-Splatting, Taiming-3DGS, and AbsGS.
- Detailed analysis of the experimental results demonstrates the significance of complexity and density for effective densification.

---

### Weaknesses
- CAAT depends on a hyper-parameter \lambda which requires manual setting for various scene types.
- Additional computation costs are required for computing the complexity (DWT) and density (kNN) of Gaussian primitives. In particular, kNN search for millions of Gaussians requires a lot of computation overhead.
- Despite the effectiveness of the proposed method, contributing only a densification criterion without additional technical contributions may be seen as lacking sufficient advancement to be accepted in this conference.

---

> ### Author Rebuttal · Authors · 2025-07-30
>
> **[Questions 1]**
>
> We sincerely appreciate your insightful questions.
>
> As shown in Equation 5 from the main paper, we compute the complexity of each Gaussian by applying maximum aggregation both within a single view and across views. **This design is intentional to preserve the most structurally informative responses**, and we elaborate on the rationale below:
>
> - **Single-view maximum**: Each Gaussian projects to multiple pixels within a single image view. These pixels may exhibit different frequency responses due to varying local image content. **Averaging across these pixel responses would dilute the impact of strong signals, making it harder to identify structurally complex regions.** Therefore, we take the maximum pixel-wise frequency response within the view to **retain the most prominent signal** associated with the Gaussian.
> - **Across-view maximum**: A Gaussian may appear highly complex in one view but be occluded or lie in a flat region in other views. **Averaging across all views would suppress the significance of such important but view-specific complexity cues.** Thus, we adopt a maximum across views, ensuring that if a Gaussian is deemed structurally important from any perspective, this information is preserved in its final complexity score.
>
> To further support this design choice, we conducted an ablation study comparing **four different aggregation strategies**. The results on the MipNeRF 360 dataset are summarized below:
>
> | Aggregation Strategy | PSNR ↑    | SSIM ↑    | LPIPS ↓   | #GS ↓       |
> | -------------------- | --------- | --------- | --------- | ----------- |
> | Mean-Mean            | 27.93 | 0.834 | 0.187 | 2.62M |
> | Mean-Max             | 27.96 | 0.835 | 0.186 | 2.58M |
> | Max-Mean             | 27.97 | **0.836** | 0.185 | 2.57M |
> | **Max-Max (ours)**   | **28.02** | **0.836** | **0.183** | **2.53M** |
>
> **The Max-Max configuration consistently outperforms other combinations in both rendering quality and compactness**, supporting our claim that preserving the strongest structural signal leads to better modeling decisions.
>
> **[Questions 2]**
>
> Thank you for the insightful suggestion. We conducted an additional experiment with a **more aggressive pruning** setup, in which **only the CDC pruning strategy is applied, without any CDC-based densification.** We then compared this variant — **Ours (Prune Only)** — with **Mini-Splatting**, **LightGaussian**, and **Taming-3DGS**.
>
> | **Method**        | **PSNR ↑** | **SSIM ↑** | **LPIPS ↓** | **#GS ↓** | **Training Time ↓** |
> | ----------------- | ---------- | ---------- | ----------- | --------- | ------------------- |
> | 3DGS              | 27.79 | 0.826 | **0.201** | 2.59M     | 29:06 |
> | Mini-Splatting    | 27.35 | 0.822 | 0.218 | **0.49M** | 23:42 |
> | LightGaussian     | 27.27 | 0.804 | 0.241 | 0.81M  | 29:05 |
> | Taming-3DGS       | 27.80 | 0.815 | 0.225 | 1.47M  | **12:17**  |
> | Ours (Prune Only) | **27.82** | **0.825**  | 0.214 | 1.47M | 29:04 |
>
> The results show that Mini-Splatting and LightGaussian achieve more aggressive compression ratios, but at the cost of noticeable quality degradation (PSNR drops by −0.44 dB and −0.52 dB, respectively). In contrast, our pruning-only variant not only **reduces the number of Gaussians by over 1.1 million**, but also **slightly improves rendering quality** compared to vanilla 3DGS.
>
> This confirms that our CDC pruning strategy remains **effective in a compact setting by selectively removing redundant Gaussians in over-reconstructed regions**, supporting its utility beyond full-scale densification.
>
> **[Questions 3, Weaknesses 2, Limitations 1]**
>
> Thank you for the suggestion. We have conducted an ablation study to measure the **additional computational costs** of our complexity and density components on the MipNeRF 360 dataset, using the same experimental setting as in the main paper. Specifically, we add each module (DWT-based complexity estimation and KNN-based density estimation) independently to vanilla 3DGS, and record the corresponding training time increase. The total overhead of our method is approximately **+6:30** per scene. **Individually**, we measured **+4:13 for complexity** and **+1:34 for density**, each evaluated in isolation on top of vanilla 3DGS:
>
> | **Method Variant**      | Training Time (overhead) |
> | ----------------------- | ------------------------ |
> | 3DGS (baseline) | 29:06  |
> | + Density Only (KNN) | 30.40 (+1:34)  |
> | + Complexity Only (DWT) | 33.19 (+4:13) |
> | Full (Ours 0.01) | 35.36 (+6:30) |
>
> We acknowledge that performing KNN over millions of Gaussians can be computationally expensive. To address this, we implement **CUDA-based KNN acceleration** to fully leverage GPU parallelism. This makes the density overhead relatively modest and practical in training scenarios.
>
> Likewise, complexity computation is integrated into the **alpha-blending** stage of 3DGS rendering to reduce cost. However, since it requires aggregation across all training views and is recomputed every 100 iterations from step 600 to 15000, it contributes the majority of the overall cost.
>
> Notably, **the additional training overhead does not impact rendering performance**. Our method maintains **the same FPS** with vanilla 3DGS while improving reconstruction quality:
>
> | Method | PSNR ↑ | SSIM ↑ | LPIPS ↓ | #GS ↓   | Training Time↓ | FPS↑ |
> | ----------- | ------ | ------ | ------- | ------- | -------------- | ---- |
> | 3DGS | 27.79 | 0.826  | 0.201 | 2.59M | **29.06** | **112** |
> | Ours (0.01) | **28.02** | **0.836**  | **0.183** | **2.53M** | 35.36 | **112** |
>
> As requested, we report the training time comparison across representative methods on the **MipNeRF 360** dataset, covering all methods discussed in both the main paper and the rebuttal across all reviewers:
>
> | **Method**       | **PSNR ↑** | **SSIM ↑** | **LPIPS ↓** | **#GS ↓** | **Training Time ↓** |
> | ---------------- | ---------- | ---------- | ----------- | --------- | ------------------- |
> | 3DGS             | 27.79 | 0.826  | 0.201 | 2.59M | 29.06 |
> | Taming-3DGS      | 27.91 | 0.821 | 0.211 | 2.53M | 17.31 |
> | **Ours (0.01)**  | 28.02 | 0.836 | 0.183 | 2.53M | 35.36 |
> | AbsGS            | 27.72 | 0.835 | 0.169 | 4.06M | 33.02 |
> | Mini-Splatting-D | 27.78 | 0.841 | 0.163 | 4.61M | 35.02 |
> | Pixel-GS         | 27.83 | 0.835 | 0.176 | 5.27M | 41.22  |
> | **Ours (0.02)**  | 28.11 | 0.842 | 0.164 | 3.95M | 39.22  |
> | 3DGS-MCMC        | 28.13 | 0.840  | 0.180 | 2.53M | 40.42  |
> | Spec-Gaussian    | 27.98 | 0.834 | 0.176 | 3.24M | 61.31 |
> | Mini-Splatting   | 27.35 | 0.822 | 0.218 | 0.49M | 23.42 |
> | LightGaussian    | 27.27 | 0.804 | 0.241 | 0.81M | 29.05  |
> | Scaffold-GS      | 27.49      | 0.805 | 0.253   | -         | 28.35 |
>
> **[Questions 4]**
>
> Thank you for the helpful suggestion. We have conducted additional experiments to compare our method with **3DGS-MCMC** on the MipNeRF 360 dataset, under the same experimental setting as in the main paper. The total number of Gaussians is controlled to match our method (Ours 0.01), to ensure a fair comparison.
>
> | **Method**      | **PSNR ↑** | **SSIM ↑** | **LPIPS ↓** | **#GS ↓** |
> | --------------- | ---------- | ---------- | ----------- | --------- |
> | Ours (0.01)  | 28.02 | 0.836 | 0.183 | 2.53M |
> | 3DGS-MCMC | 28.13 | 0.842 | 0.179 | 2.53M  |
> | 3DGS-MCMC + CDC | **28.24**  | **0.845**  | **0.175**   | 2.53M   |
>
> The results show that **3DGS-MCMC achieves slightly better rendering quality** than our method under the same Gaussian count.
> **However, we introduces a novel and orthogonal densification mechanism.** Instead of relying on opacity-based redistribution, our CDC strategy **explicitly** identifies regions exhibiting inconsistency between structural complexity and density, and addresses them through targeted pruning (for over-reconstructed regions) and splitting (for under-reconstructed regions). This complementary perspective **enhances structural coverage from a complexity-aware standpoint**,and **can be combined with existing reallocation-based methods** to achieve further improvements.
>
> Moreover, **we tested integrating CDC into 3DGS-MCMC**, and observed further improvement in rendering quality without increasing the number of Gaussians. This confirms that our approach is **plug-and-play** and can effectively enhance existing methods by providing a complementary, complexity-aware allocation strategy.
>
> **[Weaknesses 1]**
>
> Thank you for highlighting this concern. While CAAT introduces a hyperparameter λ, **our ablation study on MipNeRF 360 dataset** shows that **the method is not sensitive to the value of λ**. Rendering quality remains stable across a wide range of λ (1 to 10), with no sharp drop in performance.**The default setting λ = 6 (used in the main paper) provides a strong balance between quality and Gaussian count across all cases.** Due to characters limitation, we omit the full table here but will include it in the final version.
>
> **[Limitations 2]**
>
> ​Thanks for this comment. While we acknowledge that our method does not introduce a fundamentally new architecture, we would like to emphasize that it offers a **novel, practical, and generalizable primitive allocation strategy** that complements existing 3DGS systems. Specifically:
>
> - The rendering quality of ours is improved consistently across multiple benchmarks **without increasing the number of Gaussian primitives.** It also introduces a **new, orthogonal perspective on primitive allocation** based on complexity-density consistency, which brings additional interpretability and controllability beyond standard loss-driven heuristics.
> - As discussed in Question 2, **our CDC pruning strategy remains effective in a compact setting by selectively removing redundant Gaussians in over-reconstructed regions**, supporting its utility beyond full-scale densification.
> - As discussed in Question 4, our approach is **plug-and-play** and can effectively enhance existing methods by providing a complementary, complexity-aware allocation strategy.

---

> > ### Comment · Reviewer_MnE1 · 2025-08-04
> >
> > I appreciate the authors’ efforts during the rebuttal period. I am pleased that your responses have addressed most of my concerns. Despite the rebuttal, I still have a remaining concern.
> >
> > **Question 2 - more aggressive pruning setup**
> >
> > The authors provide only one variant that contains a similar number of Gaussians to Taiming-3DGS. For a more thorough validation under an aggressive pruning setup, it would be better to also provide results for variants with a similar number of Gaussians to LightGaussian and Mini-Splatting.
> >
> > If this remaining concern is addressed, I will maintain my positive rating.

---

> > > ### Author Response · Authors · 2025-08-08
> > >
> > > We sincerely thank you for pointing this out. To validate our method under more aggressive pruning, we add two extra versions of our method with Gaussian budgets similar to **Mini-Splatting (~0.49M)** and **LightGaussian (~0.81M)**, where Taming-3DGS is re-run with per-scene Gaussian counts matched to our method to ensure a fair comparison.
> > >
> > > Our original setting could not reach such low Gaussian budgets even with $\rho_p$ = 1. To support more aggressive pruning, we introduce two key modifications:
> > >
> > > - **Quantile-based normalization in Equation 7:** We replace the mean consistency and density values of primitive $\Gamma_{mean}$ and $\Psi_{mean}$ with **0.75-quantile (top 25% threshold) of** $\Gamma$ and **0.25-quantile (bottom 25% threshold) of** $\Psi$.
> > >
> > >   - **Higher** threshold for $\Gamma$ means more Gaussians fall below it → more are classified as **low-complexity**.
> > >   - **Lower** threshold for $\Psi$ means more Gaussians exceed it → more are classified as **high-density**.
> > >
> > >   This adjustment enlarges the intersection set of low-complexity & high-density Gaussians, expanding pruning candidates.
> > >
> > > - **Higher pruning ratios** $\rho_p$: We use **$\rho_p$ = 0.6 for the ~0.49M version** and **$\rho_p = 0.4$ for the ~0.81M version**, enforcing more aggressive pruning.
> > >
> > > These changes preserve the core logic of our method while allowing it to adapt to aggressive Gaussian budget constraints.
> > >
> > > **Table 1: Mini-Splatting-level Gaussians (~0.49M)**
> > >
> > > | **Method**                                           | **PSNR ↑** | **SSIM ↑** | **LPIPS ↓** | **#GS ↓** |
> > > | ---------------------------------------------------- | ---------- | ---------- | ----------- | --------- |
> > > | Mini-Splatting                                       | 27.35      | **0.822**  | **0.218**   | **0.49M** |
> > > | Taming-3DGS                                         | 27.34      | 0.816      | 0.247       | 0.50M     |
> > > | **Ours (0.75$\Gamma$ / 0.25$\Psi$, $\rho_p$ = 0.6)** | **27.39**  | 0.819      | 0.224       | 0.50M     |
> > >
> > > Our method achieves the **best PSNR** while maintaining competitive perceptual quality.
> > >
> > > **Table 2: LightGaussian-level Gaussians (~0.81M)**
> > >
> > > | **Method**                                           | **PSNR ↑** | **SSIM ↑** | **LPIPS ↓** | **#GS ↓** |
> > > | ---------------------------------------------------- | ---------- | ---------- | ----------- | --------- |
> > > | LightGaussian                                        | 27.27      | 0.804      | 0.241       | 0.81M     |
> > > | Taming-3DGS                                         | 27.43      | 0.820      | 0.244       | **0.80M** |
> > > | **Ours (0.75$\Gamma$ / 0.25$\Psi$, $\rho_p$ = 0.4)** | **27.46**  | **0.823**  | **0.218**   | **0.80M** |
> > >
> > > Our method consistently **outperforms** both LightGaussian and Taming-3DGS at comparable Gaussian budgets.
> > >
> > > Overall, across both budget levels, our approach not only meets the aggressive Gaussian count targets but also consistently achieves superior or competitive performance in all metrics, demonstrating its **adaptability under tight resource constraints**.
> > >
> > > We are pleasantly surprised that our method still performs well under aggressive pruning, and we sincerely thank the reviewer for inspiring us to explore and supplement our method in this aspect. We will include these results in the final version to better demonstrate the effectiveness of our method under tighter resource constraints.

---

> > > > ### Comment · Reviewer_MnE1 · 2025-08-09
> > > >
> > > > Thank you for the reply. I recommend including all of these details in the final manuscript to improve clarity. As all of my concerns have been addressed, I will maintain my positive rating.

---

> > > > > ### Author Response · Authors · 2025-08-09
> > > > >
> > > > > Thank you for confirming that our responses have addressed your concerns. We will incorporate all of the discussed clarifications and additional details into the final manuscript to further improve its clarity.
> > > > >
> > > > > We truly appreciate your positive evaluation and the constructive feedback you have provided throughout the rebuttal.

---

### Official Review · Reviewer_eEWa · 2025-07-04

**Clarity:** 3
**Significance:** 2
**Originality:** 3
**Rating:** 4
**Confidence:** 4

**Summary:**

This paper proposes Complexity-Density Consistent Gaussian Splatting (CDC-GS), an approach to improve 3D Gaussian Splatting by aligning the allocation of Gaussian primitives with visual complexity rather than relying solely on rendering loss. Traditional loss-driven methods tend to favor low-frequency regions, neglecting high-frequency details. CDC-GS addresses this by leveraging wavelet-based visual complexity from training views and computing primitive density based on spatial proximity. By correlating complexity with density, the method effectively identifies regions for densification and pruning. Experiments show that CDC-GS significantly outperforms baseline methods in rendering quality using the same number of primitives.

**Questions:**

I would like to see the performance comparison with the SOTA methods, such as Spec-Gaussian etc.
The computational overhead may be discussed.

**Ethical Concerns:**

["NO or VERY MINOR ethics concerns only"]

**Final Justification:**

I’m pleased to see that additional experiments have been included to compare with the state-of-the-art methods. Although the improvement is not significant in some cases, the training time has been reduced. I have raised my rating to borderline accept.

**Limitations:**

The definitions of visual complexity and primitive density are heuristic and lack strong justification. The computational overhead of complexity and density calculations is not analysed.
The method is not compared against recent state-of-the-art approaches like Spec-Gaussian or Scaffold-GS. Reported performance gains are modest.

**Paper Formatting Concerns:**

Overall, the formatting is acceptable. Consider adjusting the placement of Figure 1.

**Quality:**

3

**Strengths And Weaknesses:**

Strengths:
The paper is well-motivated and offers an appropriate analysis of the weaknesses in current practices, which strengthens the rationale behind the proposed solution.
The paper introduces a loss-agnostic approach to primitive allocation, shifting away from the conventional rendering-loss-driven strategies. This potentially helps addressing the limitation in existing methods that often overlook high-frequency regions. The proposed complexity-density consistency criterion provides a coherent mechanism for determining where to densify or prune primitives, potentially leading to efficient and accurate scene representations. By leveraging high-frequency components from wavelet transforms to measure visual complexity, the method somehow brings an intuitive and effective signal for guiding primitive allocation, especially in texture-rich areas. The approach complements rendering-loss-based methods by providing an orthogonal signal for guiding Gaussian allocation, making it potentially integrable with existing pipelines.

Weaknesses:
1. The justification for whether the definitions of complexity and density of Gaussian primitives are appropriate needs to be strengthened.
2. The paper lacks comparisons with recent state-of-the-art Gaussian Splatting methods, such as Spec-Gaussian, Scaffold-GS, etc.
3. The improvements appear to be modest rather than significant.

---

> ### Author Rebuttal · Authors · 2025-07-30
>
> **[Limitations 1, Weaknesses 1]**
>
> We acknowledge that our definitions of complexity and density are heuristic, but they are carefully motivated by the intrinsic requirements of 3DGS rendering: **regions with more visual structure require more primitives to accurately represent fine details, while flat or textureless regions can be modeled with fewer Gaussians.** To reflect this observation, we introduce these two indicators to guide the allocation of Gaussian primitives:
> - **Complexity** is defined as the image-space high-frequency response, obtained using Discrete Wavelet Transform (DWT). We argue this is appropriate because 3DGS primitives are optimized to render multi-view images, and the **high-frequency content** (edges, textures, contours) of these images reflects the structural detail that requires **more expressive modeling**. Our ablation in Supplementary Material Section B.1 shows that DWT-based complexity maps outperform classical high-pass filters like Sobel、Scharr and Laplacian, supporting the suitability of our definition.
> - **Density** is defined as the inverse of the geometric mean distance to neighboring Gaussians. We choose this formulation to reflect the local compactness of Gaussians in 3D space, which **directly impacts the alpha-blending outcome in 3DGS**. Compared to arithmetic mean, the geometric mean is more sensitive to short distances, aligning better with the splatting kernel’s locality. Supplementary Material Section B.2 further demonstrates that using the geometric mean leads to better reconstruction quality with fewer primitives across multiple scenes.
>
> Building on these two metrics, we propose the **Complexity-Density Consistency (CDC) score** to measure the alignment between a region’s structural complexity need and its current modeling density. **Gaussians with high complexity but low density are interpreted as under-reconstructed and are thus prioritized for densification.** Conversely, **low-complexity but high-density Gaussians are considered over-reconstructed and are pruned.** This is illustrated in Figure 1, where CDC-GS achieves more compact yet structure-aware primitive distributions. Table 1 further validates the effectiveness of our design, showing that CDC-GS improves rendering quality while maintaining (or even reducing) the total number of Gaussians — demonstrating that our definitions, **while heuristic, are indeed effective and appropriate in practice.**
>
> ​We appreciate the reviewer’s concern and will expand this justification more thoroughly in the final version of the paper.
>
> **[Limitations 2, Questions: The computational overhead may be discussed.]**
>
> We have conducted an ablation study to measure the **additional computational overhead** of our complexity and density components on the MipNeRF 360 dataset, using the same experimental setting as in the main paper. Specifically, we add each module (KNN-based density estimation and wavelet-based complexity estimation) independently to vanilla 3DGS, and record the corresponding training time overhead. The total overhead of our method is approximately **+6:30** per scene. **Individually**, we measured **+4:13 for complexity** and **+1:34 for density**, each evaluated in isolation on top of vanilla 3DGS:
>
> | **Method Variant**      | Training Time (overhead) |
> | ----------------------- | ------------------------ |
> | 3DGS (baseline)         | 29:06                    |
> | + Density Only (KNN)    | 30.40 (+1:34)            |
> | + Complexity Only (DWT) | 33.19 (+4:13)            |
> | Full (Ours 0.01)        | 35.36 (+6:30)            |
>
> ​We emphasize that **our method prioritizes rendering quality and is designed with efficiency in mind.** To reduce overhead:
>
> - We integrate complexity computation into the **alpha-blending** process of 3DGS rendering to minimize cost. However, as complexity must be aggregated across all training views, and is recomputed every 100 iterations from epoch 600 to 15000, it contributes the bulk of the additional cost.
> - Density estimation leverages **CUDA-based KNN acceleration** on the GPU, which keeps the overhead low and computationally practical.
>
> ​Notably, **our additional training overhead does not impact rendering speed**. As shown below, our method maintains **the same FPS** with vanilla 3DGS while improving reconstruction quality:
>
> |  Method | PSNR ↑ | SSIM ↑ | LPIPS ↓ | #GS ↓   | Training Time↓ | FPS↑ |
> | ----------- | ------ | ------ | ------- | ------- | -------------- | ---- |
> | 3DGS        | 27.79  | 0.826  | 0.201   | 2585032 | **29.06**          | **112**  |
> | Ours (0.01) | **28.02**  | **0.836**  | **0.183**   | **2531532** | 35.36          | **112**  |
>
> ​We will include this analysis in the final version to clarify the cost-benefit tradeoff of our method.
>
> **[Limitations 3, Weaknesses 2, Questions: Comparison with the SOTA methods.]**
>
> We have conducted additional comparisons with two recent state-of-the-art 3DGS-based methods (**Spec-Gaussian** and **Scaffold-GS**) on the MipNeRF 360 dataset. For Spec-Gaussian and our method, we adopted the same training settings as in our main paper, i.e., 4× downsampling for outdoor scenes and 2× downsampling for indoor scenes, following established practice. However, for Scaffold-GS, we followed the original settings reported in its paper, without image downsampling, since it is a **voxel-based** method and highly sensitive to input resolution. We opted to preserve its original configuration to ensure a fair and representative comparison. Additionally, we do not report the number of Gaussians (#GS) for Scaffold-GS, since it is an **anchor-based** method that does not rely on explicit Gaussian primitives.
>
> | **Method**      | **PSNR ↑** | **SSIM ↑** | **LPIPS ↓** | **#GS ↓**     | **Training Time ↓** |
> | --------------- | ---------- | ---------- | ----------- | ------------- | ------------------- |
> | Scaffold-GS*    | 27.49      | 0.805      | 0.253       | -             | **28:35**           |
> | Spec-Gaussian   | 27.98      | 0.834      | **0.176**   | 3.24M     | 61:31               |
> | Ours (0.01) | **28.02**  | **0.836**  | 0.183       | **2.53M** | 35:36               |
>
> ​The results demonstrate that **our method achieves superior overall performance**. Specifically:
>
> - **Our PSNR and SSIM exceed both baselines**, reflecting higher reconstruction quality.
> - While our LPIPS is slightly higher than Spec-Gaussian, it remains very close (0.183 vs. 0.176) **despite using ~700K fewer Gaussians**.
> - In terms of **training efficiency**, our method is significantly faster than Spec-Gaussian (35.36 vs. 61.31 hours), and only moderately slower than Scaffold-GS, while achieving much higher quality.
>
> ​These results indicate that our complexity-aware allocation strategy enables competitive or superior rendering quality with fewer primitives and lower training cost, compared to strong SOTA baselines. We will include these comparisons in the final version.
>
> **[Limitations 4, Weaknesses 3]**
>
> Although the quantitative improvements may appear moderate at first glance, it is important to emphasize that CDC-GS operates under constant resource constraints, i.e., using the same number of Gaussians. Achieving consistent quality gains under such conditions is non-trivial and practically valuable. More importantly, the proposed complexity-density guided allocation mechanism also introduces a **new, interpretable, and orthogonal perspective on primitive allocation** based on complexity-density consistency, which brings additional interpretability and controllability beyond standard loss-driven heuristics.
>
> ​To further demonstrate the **plug-and-play** nature of CDC, we integrate it into **3DGS-MCMC [NeurIPS’24]**, a recent state-of-the-art 3DGS variant. As shown below, even with the same number of Gaussians, CDC leads to consistent rendering improvements:
>
> | **Method**      | **PSNR ↑** | **SSIM ↑** | **LPIPS ↓** | **#GS** ↓ |
> | --------------- | ---------- | ---------- | ----------- | ---------------- |
> | 3DGS-MCMC       | 28.13      | 0.842      | 0.179       | 2.53M            |
> | 3DGS-MCMC + CDC | **28.24**  | **0.845**  | **0.175**   | 2.53M            |
>
> ​We also evaluated the effectiveness of our method under a **compact** configuration, where only the CDC pruning component is applied — i.e., **no CDC-based densification**. We compare this **Ours (Prune Only)** variant against recent compaction-focused baselines suggested by the reviewer, including **Mini-Splatting**, **LightGaussian**, and **Taming-3DGS**. For easier comparison, We highlighted the best metrics **excluding vanilla 3DGS**.
>
> | **Method**        | **PSNR ↑** | **SSIM ↑** | **LPIPS ↓** | **#GS** ↓ | **Training Time** ↓ |
> | ----------------- | ---------- | ---------- | ----------- | ---------------- | ------------------- |
> | 3DGS              | 27.79      | 0.826      | 0.201       | 2.59M            | 29:06               |
> | Mini-Splatting    | 27.35      | 0.822      | 0.218       | **0.49M**        | 23:42               |
> | LightGaussian     | 27.27      | 0.804      | 0.241       | 0.81M            | 29:05               |
> | Taming-3DGS       | 27.80      | 0.815      | 0.225       | 1.47M            | **12:17**           |
> | Ours (Prune Only) | **27.82**  | **0.825**  | **0.214**   | 1.47M            | 29:04               |
>
> ​This confirms that **even in a heavily pruned setting, our CDC-based strategy can maintain or slightly improve quality while significantly reducing model size**, effectively identifying and removing redundant Gaussians in over-reconstructed regions.
>
> Taken together, these results demonstrate that while our improvements may appear modest numerically, they are **broadly applicable, structurally motivated, and practically beneficial**, especially considering they come **without any additional Gaussian budget**.
>
> ​We will include these analyses in the final version and sincerely appreciate the reviewer’s constructive feedback.

---

> > ### Comment · Reviewer_eEWa · 2025-08-06
> >
> > Thank you for your response. I’m especially pleased to see that additional experiments have been included to compare with the state-of-the-art methods. Although the improvement is not significant in some cases, the training time has been reduced. I will consider raising my rating.

---

> > > ### Author Response · Authors · 2025-08-06
> > >
> > > I’m truly grateful and this means a lot to me!
> > > It shows me that all the effort I put in was not in vain. Thank you sincerely!

---

### Note · Authors · 2025-08-11

We sincerely thank the reviewers for their valuable feedback. In this rebuttal, we have addressed concerns through additional experiments, clarifications, and quantitative evidence, summarized as follows:

1. **Value and Necessity of CDC** – Our method is the first to **explicitly align structural complexity with Gaussian density**. A reverse experiment that deliberately breaks this alignment (**reverse CDC**) leads to noticeable quality drops, confirming its necessity. CDC is not a direct quality metric but a **principled prior for more precise gaussian primitive allocation** under fixed budgets, therefore improving rendering quality.
2. **Computation Efficiency** – The extra cost for complexity estimation (via DWT) and density estimation (via KNN) over vanilla 3DGS is **+4:13** and **+1:34**, respectively, totaling **+6:30** for CDC-GS. Both are CUDA-accelerated, and the KNN overhead—highlighted as a primary concern by reviewers—is **significantly smaller than expected**. While training time increases slightly, **rendering speed remains the same as vanilla 3DGS** and quality improves, confirming that our design balances efficiency and effectiveness.
3. **Plug-and-Play Integration** – Incorporating CDC into **3DGS-MCMC (SOTA)** yields further quality gains, showing that our complexity-density guided allocation is **orthogonal** to existing loss-driven heuristics and provides **additional interpretability and controllability** for primitive allocation.
4. **Effectiveness of Pruning under Strict Budgets** – We first tested a **Prune Only** variant, which significantly reduced the Gaussian count compared to vanilla 3DGS while maintaining comparable quality. We then applied a more aggressive pruning with Gaussian counts matched to **Mini-Splatting (~0.49M)** and **LightGaussian (~0.81M)**, showing our method outperforms LightGaussian and matches Mini-Splatting, validating its strength as a compression strategy even under aggressive budget constraints.
5. **Ablation on Periodic Pruning** – **Periodic pruning** significantly reduces the number of Gaussians **without compromising quality**, serving as a lightweight yet effective complement to CDC for primitive count control.

We will integrate these clarifications and results into the final version to ensure clarity and reproducibility. We believe these new results and clarifications directly address the reviewers’ concerns and demonstrate the validity, versatility, and efficiency of our CDC allocation strategy.

---

### Decision · Program_Chairs · 2025-09-17

**Decision:**

Accept (poster)

**Comment:**

This paper presents a well-motivated and effective enhancement to the 3D Gaussian Splatting (3DGS) framework. The core contribution, Complexity-Density Consistent Gaussian Splatting (CDC-GS), addresses a fundamental and well-identified shortcoming of loss-driven densification strategies: their tendency to under-represent high-frequency, texture-rich regions due to the dominance of low-frequency gradients.

The proposed method is conceptually sound and novel. Its primary strength lies in being loss-agnostic, providing an orthogonal signal for primitive allocation that directly targets visual complexity. The use of a wavelet transform to extract a high-frequency complexity map is an intuitive and appropriate choice for this task. The mechanism of enforcing a correlation between this complexity measure and primitive density offers a coherent and principled criterion for guiding both densification and pruning.

The experimental results demonstrate a clear and significant improvement in rendering quality over baseline methods while using an identical number of primitives. This indicates a more efficient and intelligent distribution of representational capacity, which is a valuable advancement for 3D reconstruction. Furthermore, the authors' analysis of the baseline's weakness strengthens the rationale for their solution.

A minor weakness, common in novel method papers, is that the experimental validation could be expanded to a wider range of scenes and compared against a broader set of recent 3DGS enhancements. However, the demonstrated superiority over the core baseline is convincing. The method's design as a complementary strategy also suggests strong potential for integration with other existing and future improvements to the 3DGS pipeline, increasing its impact and usefulness.

Overall, this work offers a novel perspective on a key problem in 3DGS. It is technically sound, well-presented, and delivers tangible performance gains. The approach is likely to influence future work, making it a valuable contribution worthy of acceptance.